# Polygenic signals of sex differences in selection in humans from the UK Biobank

**Filip Ruzicka**[1]*, **Luke Holman**[2,3], **Tim Connallon**[1]

**1** School of Biological Sciences, Monash University, Clayton, Victoria, Australia, **2** School of BioSciences, University of Melbourne, Parkville, Victoria, Australia, **3** School of Applied Sciences, Edinburgh Napier University, Edinburgh, United Kingdom

* Filip.Ruzicka@monash.edu

**Data Availability Statement:** All relevant code is available on the following public github repositories (/filipluca/polygenic_SA_selection_in_the_UK_biobank/ and /lukeholman/UKBB_LDSC/) and all

## Abstract

Sex differences in the fitness effects of genetic variants can influence the rate of adaptation and the maintenance of genetic variation. For example, "sexually antagonistic" (SA) variants, which are beneficial for one sex and harmful for the other, can both constrain adaptation and increase genetic variability for fitness components such as survival, fertility, and disease susceptibility. However, detecting variants with sex-differential fitness effects is difficult, requiring genome sequences and fitness measurements from large numbers of individuals. Here, we develop new theory for studying sex-differential selection across a complete life cycle and test our models with genotypic and reproductive success data from approximately 250,000 UK Biobank individuals. We uncover polygenic signals of sex-differential selection affecting survival, reproductive success, and overall fitness, with signals of sex-differential reproductive selection reflecting a combination of SA polymorphisms and sexually concordant polymorphisms in which the strength of selection differs between the sexes. Moreover, these signals hold up to rigorous controls that minimise the contributions of potential confounders, including sequence mapping errors, population structure, and ascertainment bias. Functional analyses reveal that sex-differentiated sites are enriched in phenotype-altering genomic regions, including coding regions and loci affecting a range of quantitative traits. Population genetic analyses show that sex-differentiated sites exhibit evolutionary histories dominated by genetic drift and/or transient balancing selection, but not long-term balancing selection, which is consistent with theoretical predictions of effectively weak SA balancing selection in historically small populations. Overall, our results are consistent with polygenic sex-differential—including SA—selection in humans. Evidence for sex-differential selection is particularly strong for variants affecting reproductive success, in which the potential contributions of nonrandom sampling to signals of sex differentiation can be excluded.

## Introduction

Adaptation of a population to its environment requires heritable genetic variation for fitness [1]. Although many populations show substantial genetic variation for fitness components [2]

relevant data is available within the manuscript, Supporting Information files, and at https://zenodo. org/record/6824671.

**Funding:** This work was supported by an Australian Research Council Discovery Project Grant FT170100328, to TC. (www.arc.gov.au) The funders had no role in study design, data collection and analysis, decision to publish, or preparation of the manuscript.

**Competing interests:** The authors have declared that no competing interests exist.

**Abbreviations:** FDR, false discovery rate; GWAS, genome-wide association study; LD, linkage disequilibrium; LRS, lifetime reproductive success; NCD, non-central deviation; SA, sexually antagonistic; SC, sexually concordant; SHBG, sex hormone binding globulin; SNP, single-nucleotide polymorphism.

—including life history traits such as maturation rate, lifespan, mating success, and fertility [2,3]—genetic trade-offs between components or between different types of individuals in a population, limit adaptive potential [4]. For example, a mutation that increases the probability of survival to adulthood might simultaneously decrease adult reproductive success (e.g., [5]), weakening the mutation's net fitness effect [4]. In addition to slowing adaptation [6–8], genetic trade-offs can increase standing genetic variation [2,9], give rise to balancing selection [10,11], and favour evolutionary transitions between mating systems [12,13], modes of sex determination [14], and genome structures [15–18].

Sexually antagonistic (SA) genetic polymorphisms—in which the alleles that benefit one sex are harmful to the other—are a type of genetic trade-off that may be common in sexually reproducing species [19]. Theory shows that SA polymorphisms are likely to arise when mutations differentially affect trait expression in each sex or when mutations similarly affect traits under divergent directional selection between the sexes [20]. Empirical quantitative genetic studies imply that both conditions are frequently met in nature [21–24] and, accordingly, that SA polymorphisms contribute to phenotypic variation in a range of plant and animal populations (e.g., [25–27]), including humans [28–31].

Although there is now abundant evidence that SA polymorphisms contribute to phenotypic variation, efforts to identify and characterise SA alleles in genomic data face 2 formidable challenges [32]. First, methods using explicit fitness measurements to identify SA polymorphisms (e.g., genome-wide association studies (GWAS) of fitness [33]) are rarely feasible, because it is challenging to obtain fitness measurements for large numbers of genotyped individuals under natural conditions [2]. Second, methods using allele frequency differences between adult females and males as genomic signals of SA viability selection (e.g., between-sex $F_{ST}$ estimates [32,34–43]) are limited in several ways: They have low power to detect SA loci, they cannot distinguish SA selection from sex differences in the strength of selection, they are susceptible to artefacts generated by population structure and mis-mapping of sequence reads to sex chromosomes [32,40,41,44], and they neglect fitness components other than viability, such as reproductive success [32,45]. Previous studies of human genomic data [32,34–36,43,44,46] have been affected by one or more of these issues, such that we currently lack robust evidence of SA genomic variation in humans. More generally, these impediments help to explain the limited catalogue of SA polymorphisms across species [47–49], which currently comprises a handful of loci with exceptionally large phenotypic effects (e.g., [50–54]).

Despite these challenges, new datasets and analytical approaches provide opportunities to identify robust genomic signals of SA selection. First, massive "biobank" datasets, which are widely used in human genomics, sometimes include both genotype and offspring number data [29,55] that can be used to detect loci with SA effects on reproductive components of fitness [32]. Second, estimates of allele frequency differences between sexes—though ill-suited for confidently identifying individual SA loci affecting viability—may nevertheless be amenable to genome-wide tests for polygenic SA viability selection [32,34]. Third, population genomic metrics of sex-differential selection (e.g., between-sex $F_{ST}$) may include an appreciable proportion of genuine SA loci in the upper tails of their distributions, providing a set of candidate loci that can collectively yield insights into the general properties of SA polymorphisms (e.g., their functional characteristics and evolutionary dynamics), despite uncertainty about individual candidates.

Here, we extend [32,34] and develop new statistical tests based on $F_{ST}$ metrics of between-sex allele frequency differentiation to detect polygenic signals of sex-differential selection affecting viability, reproduction, and total fitness during a full generational cycle. Applying these tests to the UK Biobank [55]—a dataset comprising quality-filtered genotype and offspring number data for approximately 250,000 men and women—reveals polygenic signals of sex-differential and SA polymorphism. We corroborate these results by using mixed-model

statistics that explicitly control for systematic differences in the genetic ancestry of female and male individuals. We minimise potential sequencing artefacts and further show that sex-differentiated polymorphisms are preferentially situated in functional, phenotype-altering genomic sequences. Finally, we use genetic diversity data to examine modes of evolution affecting sex-differentiated sites.

## Results

### Genomic signals of sex differences in selection: Theoretical predictions

Previous studies have examined sex-differential effects of genetic variation during the zygote-to-adult stage by comparing allele frequencies between adult females and males [32,34,36–40,44]. By contrast, our analytical approach combines allele frequency with offspring number data to estimate sex-differential effects during a full generational life cycle (Fig 1). To illustrate

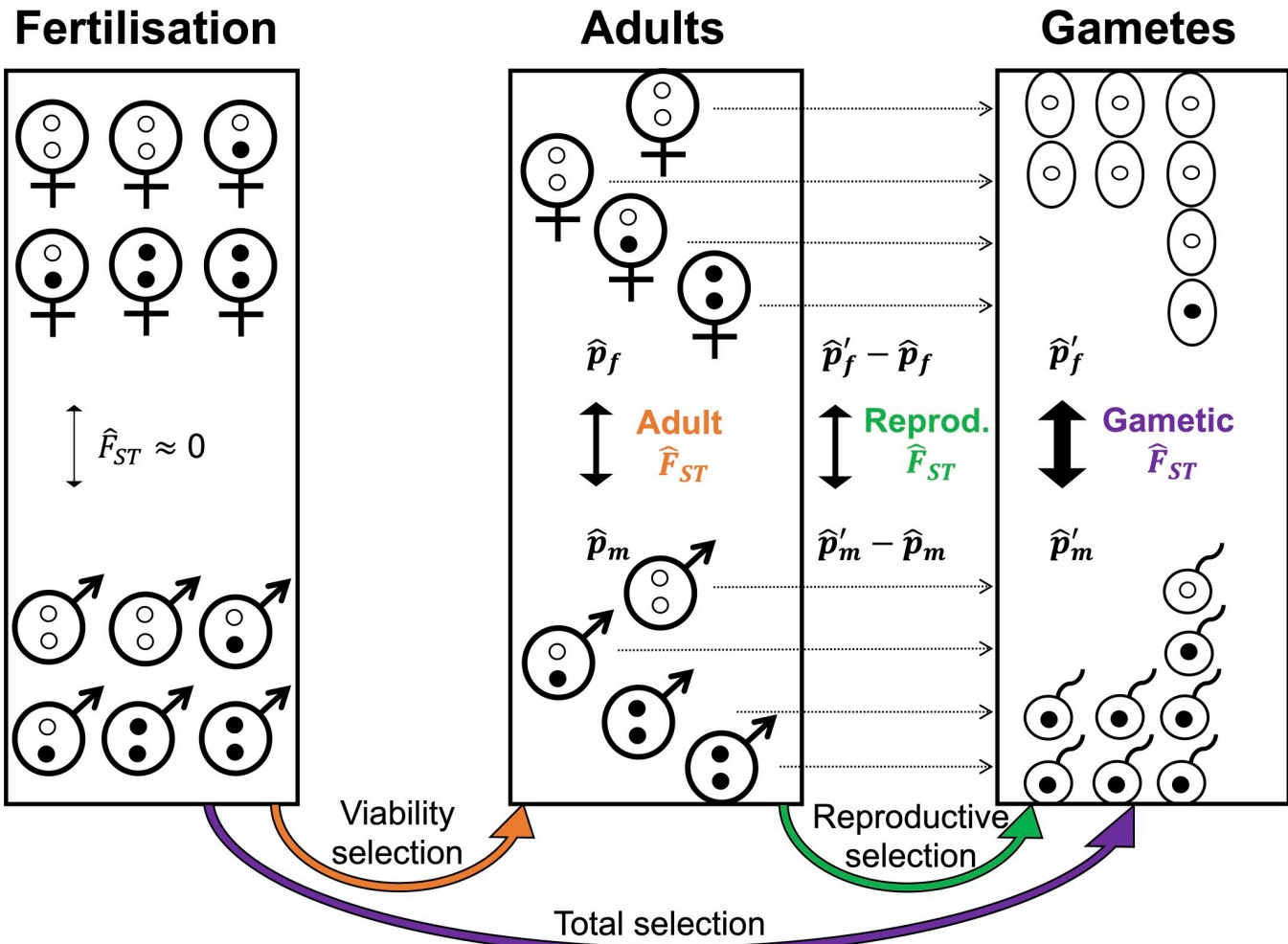

**Fig 1. Partitioning signals of sex differences in selection among fitness components.** A pair of autosomal alleles are represented by white and black dots, representing female- and male-beneficial alleles, respectively; $\hat{p}_f$, $\hat{p}_m$, $\hat{p}'_f$, and $\hat{p}'_m$ depict sex-specific frequency estimates for a given allele at different stages of the life cycle (see main text for details). Autosomal allele frequencies are equalised between sexes at fertilisation (left box; females, top; males, bottom), resulting in negligible allele frequency differentiation at this stage of the life cycle. Differentiation between sexes can arise in the sample of adults (middle box) due to sex differences in viability selection among juveniles (orange arrow) and in the projected gametes (right box) due to sex differences in LRS among adults (green arrow). Data on sex-specific allele frequencies and LRS thus allow the estimation of sex-differential effects of genetic variants on each fitness component (including overall fitness; purple arrow), despite the absence of allele frequency data among zygotes (left box) and gametes (right box), which are inferred and not directly observed. LRS, lifetime reproductive success.

the approach, consider a large, well-mixed population containing many polymorphic, biallelic, autosomal loci. At fertilisation, mendelian inheritance equalises allele frequencies between the sexes (Fig 1, left box). In the zygote-to-adult stage, loci with sex-differential effects on survival accumulate allele frequency differences between the adults of each sex (e.g., the black allele becomes enriched in adult males and deficient in adult females because it improves zygote-to-adult survival in males but reduces it in females; Fig 1, middle box). Among the adults, alleles with sex-differential effects on reproductive success have different transmission rates to the next generation from surviving females versus surviving males (e.g., the black allele is enriched among the male gametes contributing to fertilisation but deficient among female gametes, thus increasing its transmission to offspring of males but decreasing transmission to offspring of females; Fig 1, right box).

Adult allele frequencies, coupled with offspring number data per individual, thus provide an opportunity to estimate sex-differential effects of genetic variation during a complete life cycle, even though zygotic and gametic allele frequencies are inferred and not directly observed. Below, we apply our approach to the UK Biobank, a dataset that includes genotypes and reported offspring numbers (hereafter "lifetime reproductive success" or LRS, following standard terminology [29]) among putatively post-reproductive adults (ages 45 to 69 after filtering; see Materials and methods). For a biallelic autosomal locus with alleles $A_1$ and $A_2$, we denote $\hat{p}_m$ and $\hat{p}_f$ the respective estimated frequencies of the $A_1$ allele in adult males and females of the UK Biobank. The projected frequencies of $A_1$ in paternal and maternal gametes contributing to fertilisation are:

$$\hat{p}'_m = \frac{M_{11} + \frac{1}{2}M_{12}}{M_{11} + M_{12} + M_{22}} \tag{1A}$$

$$\hat{p}'_f = \frac{F_{11} + \frac{1}{2}F_{12}}{F_{11} + F_{12} + F_{22}} \tag{1B}$$

where $M_{ij}$ and $F_{ij}$ represent the cumulative LRS of males and females, respectively, with genotype $ij$ (e.g., $M_{11}$, $M_{12}$, and $M_{22}$ correspond to genotypes $A_1A_1$, $A_1A_2$, and $A_2A_2$).

Using $F_{ST}$ [56], we partition between-sex allele frequency differentiation over 1 generation into 3 components: (i) differentiation among adults, which includes effects of sex-differential survival (hereafter "adult $F_{ST}$;" see [32,34,45]); (ii) sex-differential variation in adult LRS (hereafter "reproductive $F_{ST}$"); and (iii) sex-differential variation in overall fitness (hereafter "gametic $F_{ST}$"). Single-locus estimates of adult, reproductive, and gametic $F_{ST}$ are defined, respectively, as:

$$\hat{F}_{ST(Adult)} = \frac{(\hat{p}_f - \hat{p}_m)^2}{4\bar{p}(1 - \bar{p})} \tag{2A}$$

$$\hat{F}_{ST(Reproductive)} = \frac{((\hat{p}'_f - \hat{p}_f) - (\hat{p}'_m - \hat{p}_m))^2}{4\bar{p}(1 - \bar{p})} \tag{2B}$$

$$\hat{F}_{ST(Gametic)} = \frac{(\hat{p}'_f - \hat{p}'_m)^2}{4\bar{p}'(1 - \bar{p}')} \tag{2C}$$

where $\bar{p} = (\hat{p}_f + \hat{p}_m)/2$ and $\bar{p}' = (\hat{p}'_f + \hat{p}'_m)/2$.

**$F_{ST}$ distributions in the absence of sex-differential selection.** In the absence of sex differences in selection (e.g., under neutrality or under sexually concordant (SC) selection of equal

magnitude and direction in each sex), with large sample sizes, negligible Hardy–Weinberg deviations at birth, and excluding single-nucleotide polymorphisms (SNPs) with very low minor allele frequencies, we show that the adult, reproductive, and gametic $\hat{F}_{ST}$ metrics converge, respectively, to the following distributions:

$$\hat{F}_{ST(Adult)} \approx \left( \frac{1}{8N_f} + \frac{1}{8N_m} \right) X \tag{3A}$$

$$\hat{F}_{ST(Reproductive)} \approx \frac{\frac{\hat{p}_f(1-\hat{p}_f)}{2N_f} \frac{\sigma_f^2}{\mu_f^2} \left( 1 - \hat{F}_{IS}^f \right) + \frac{\hat{p}_m(1-\hat{p}_m)}{2N_m} \frac{\sigma_m^2}{\mu_m^2} \left( 1 - \hat{F}_{IS}^m \right)}{4\hat{p}(1-\hat{p})} X \tag{3B}$$

$$\hat{F}_{ST(Gametic)} \approx \left[ \frac{1}{8N_f} \left( 1 + \frac{\sigma_f^2}{\mu_f^2} \right) + \frac{1}{8N_m} \left( 1 + \frac{\sigma_m^2}{\mu_m^2} \right) \right] X \tag{3C}$$

where each $X$ is an independent chi-square random variable with 1 degree of freedom, $N_f$ and $N_m$ denote adult sample sizes, $\mu_f$ and $\mu_m$ denote mean LRS, $\sigma_f^2$ and $\sigma_m^2$ denote variances in LRS, and $\hat{F}_{IS}^f$ and $\hat{F}_{IS}^m$ quantify sex-specific departures from Hardy–Weinberg equilibrium in the sample of adults (Section A in S1 Appendix). In datasets such as the UK Biobank, there is also between-site variation in the number of genotyped individuals and the extent of Hardy–Weinberg deviations in the adult sample. The null distributions described by Eqs [3A–3C] are easily adjusted to account for this between-site variation (see Materials and methods).

Relative to the null distributions in Eqs [3A–3C], sex differences in selection inflate each $\hat{F}_{ST}$ metric (Section A in S1 Appendix). These inflations may arise due to polymorphisms under sex-differential selection and neutral polymorphisms that hitchhike with selected polymorphisms. However, linkage disequilibrium (LD) alone cannot inflate genome-wide $\hat{F}_{ST}$ in the absence of genuine selected polymorphisms (Section B in S1 Appendix). As such, $\hat{F}_{ST}$ inflations represent reliable signals of sex-differentially selected polymorphism [32], provided: (i) technical artefacts are controlled (as shown below); (ii) sex-specific population structure is controlled; and (iii) males and females are sampled at random (though (iii) is not a requirement for reproductive $\hat{F}_{ST}$; see Discussion). To simplify the presentation, we first present analyses using $F_{ST}$ metrics, but we return to non-$F_{ST}$ metrics in the section titled "Controlling for sex-specific population structure."

## Genomic signals of sex differences in selection: Empirical data

**UK Biobank SNP data.** The sample size in the UK Biobank, after removing individuals that were closely related, had a recorded ancestry other than "White British," or had missing LRS data, was $N = 249,021$ ($N_m = 115,531$ males and $N_f = 133,490$ females). We removed rare polymorphic sites (MAF < 1%), sites with low genotype or imputation quality, and sites with high potential for artefactual between-sex differentiation based on criteria identified by Kasimatis and colleagues [44] (i.e., between-sex differences in missing rates, deficits of minor allele homozygotes, and heterozygosity levels exceeding what can be plausibly be explained by sex differences in selection; see Section C in S1 Appendix). Reassuringly, none of the 8 sites that Kasimatis and colleagues [44] identified as false positives for sex-differential viability selection appear among the quality-filtered, LD-pruned, imputed SNPs ($N = 1,051,949$) that are the focus of our analyses.

**Observed $F_{ST}$ distributions relative to null distributions.** We tested for sex differences in selection by calculating adult, reproductive, and gametic $\hat{F}_{ST}$ (Eqs [2A–2C]) in

the UK Biobank and contrasting these estimates against: (i) their respective theoretical null distributions (Eqs [3A–3C]); and (ii) empirical null distributions (generated by a single random permutation of male and female labels among individuals or, in the case of reproductive $\hat{F}_{ST}$, a single permutation of LRS among individuals of each sex; see Materials and methods).

All 3 $\hat{F}_{ST}$ metrics showed greater between-sex differentiation than predicted by their theoretical and empirical null distributions, consistent with sex differences in selection with respect to mortality, LRS, and total fitness. Mean adult $\hat{F}_{ST}$ in the observed data was larger than predicted by both null distributions (theoretical null: $2.039 \times 10^{-6}$; permuted null: $2.043 \times 10^{-6}$; observed: $2.104 \times 10^{-6}$; Wilcoxon and Kolmogorov–Smirnov tests, $p < 0.001$; Fig 2A and 2D), with a 14.1% and 13.7% excess of SNPs in the top percentile of the theoretical and empirical nulls, respectively ($\chi^2$ tests, $p < 0.001$). Mean reproductive $\hat{F}_{ST}$ was also larger than predicted by both nulls (theoretical null: $8.731 \times 10^{-7}$; permuted null: $8.749 \times 10^{-7}$; observed: $8.900 \times 10^{-7}$; Wilcoxon and Kolmogorov–Smirnov tests, $p < 0.001$; Fig 2B and 2E), with a 7.4% and 5.0% excess of SNPs in the top percentile of the theoretical and empirical nulls ($\chi^2$ tests, $p < 0.001$). Moreover, mean gametic $\hat{F}_{ST}$ was larger than predicted by both nulls (theoretical null: $2.908 \times 10^{-6}$; permuted null: $2.907 \times 10^{-6}$; observed: $2.974 \times 10^{-6}$; Wilcoxon and Kolmogorov–Smirnov tests, $p < 0.001$; Fig 2C and 2F), with a 9.0% and 7.8% excess of SNPs in the top percentile of the theoretical and empirical nulls ($\chi^2$ tests, $p < 0.001$).

Signals of sex differences in selection in adult, reproductive, and gametic $\hat{F}_{ST}$ were polygenic. For example, genetic variants situated in genomic regions with high LD tended to explain more SNP heritability of each $\hat{F}_{ST}$ metric than variants situated in low-LD regions, as predicted if each sex-differential fitness component has a polygenic basis (Section D in S1 Appendix). Moreover, no individual locus had a $p$-value below the Bonferroni-corrected threshold of $4.753 \times 10^{-8}$, implying that the significant overall inflations were not driven by a small number of strongly sex-differentiated polymorphisms (adult $\hat{F}_{ST}$: minimum p- and q-values = $2.237 \times 10^{-7}$ and 0.176; reproductive $\hat{F}_{ST}$: minimum p- and q-values = $3.925 \times 10^{-7}$ and 0.413; gametic $\hat{F}_{ST}$: minimum p- and q-values = $4.152 \times 10^{-6}$ and 0.821).

## Forms of sex-differential selection: Theoretical predictions

The $\hat{F}_{ST}$ elevations reported above indicate the presence of polygenic sex-differential selection in the UK Biobank. However, the signals could have arisen because of SA selection, because of sex differences in the strength but not the direction of selection (i.e., sex-differential SC selection), or a combination of both scenarios. To partition signals affecting LRS into SA and SC components, we examined the effects of a given allele on LRS in each sex relative to the other. Specifically, estimates of the product $(\hat{p}'_m - \hat{p}_m)(\hat{p}'_f - \hat{p}_f)$ should tend to be negative when alleles have SA effects and positive when alleles have SC effects (Fig 3A). A new metric, termed "unfolded reproductive $\hat{F}_{ST}$," provides a standardised measure of the product of sex-specific effects on LRS:

$$\hat{F}_{ST(Unfolded)} = \frac{(\hat{p}'_m - \hat{p}_m)(\hat{p}'_f - \hat{p}_f)}{\sqrt{\frac{\hat{p}_m(1-\hat{p}_m)}{2N_m}\frac{\sigma_m^2}{\mu_m^2}\left(1 - \hat{F}_{IS}^m\right)\frac{\hat{p}_f(1-\hat{p}_f)}{2N_f}\frac{\sigma_f^2}{\mu_f^2}\left(1 - \hat{F}_{IS}^f\right)}} \tag{4}$$

In the absence of any selection on LRS, unfolded reproductive $\hat{F}_{ST}$ is distributed as the product of 2 independent, standard normal distributions (i.e., symmetrically distributed with a mean of

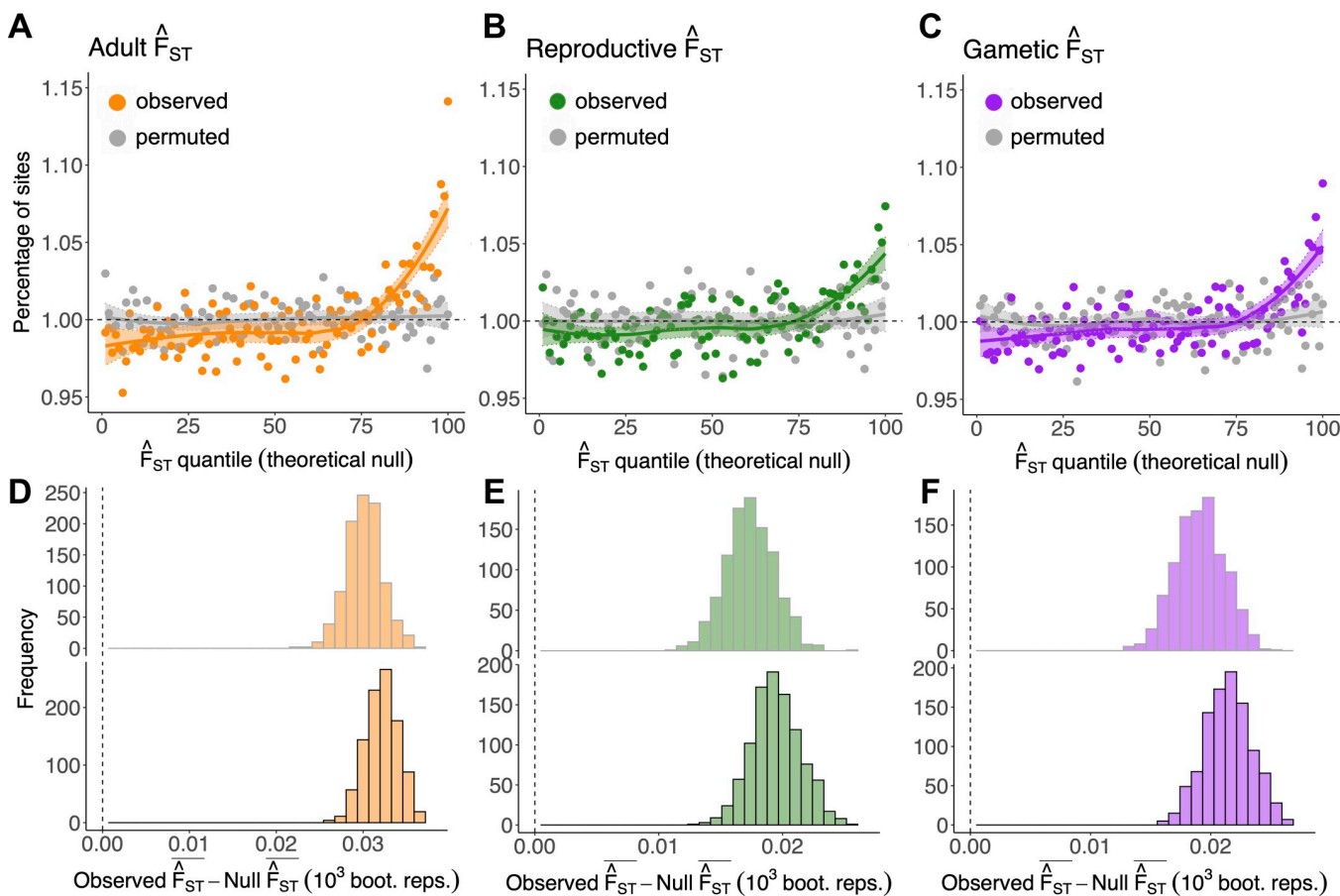

**Fig 2.** Polygenic signals of sex-differential selection: Inflation in $\hat{F}_{ST}$ metrics relative to their nulls. (**A–C**) Percentage of sites (coloured, observed; grey, permuted) falling into each of 100 quantiles of the theoretical null distributions of adult $\hat{F}_{ST}$ (A), reproductive $\hat{F}_{ST}$ (B), and gametic $\hat{F}_{ST}$ (C). Theoretical null data (x-axes) were generated by simulating values (nSNPs = 1,051,949) from a chi-square distribution with 1 degree of freedom. For each locus, observed and permuted $\hat{F}_{ST}$ values were scaled by the multiplier of the relevant theoretical null distributions (i.e., the multiplier in Eqs [3A–3C] for adult, reproductive, and gametic $\hat{F}_{ST}$, respectively; see Materials and methods). In the absence of sex differences in selection, approximately 1% of observed SNPs should fall into each quantile of the null (dashed line). LOESS curves (±SE) are presented for visual emphasis. (**D–F**) Difference between the mean of observed and empirical null data for each metric (i.e., adult, reproductive, and gametic $\hat{F}_{ST}$, respectively) (top), and the difference between observed and theoretical null data (bottom), across 1,000 bootstrap replicates. Vertical line intersects zero (no difference between observed and null data). As in panels (A–C), $\hat{F}_{ST}$ values were scaled by the relevant theoretical null distributions. The code and data needed to generate this figure can be found at https://github.com/filipluca/polygenic_SA_selection_in_the_UK_biobank and https://zenodo.org/record/6824671. SNP, single-nucleotide polymorphism.

zero; see Section E in S1 Appendix). SA selection generates an excess of loci in the lower quantiles of this null model, while SC selection generates an excess of loci in the upper quantiles of the null. Note that sex differences in SC selection are not required to generate an excess of positive values for unfolded reproductive $\hat{F}_{ST}$ (SC selection of equal magnitude in the sexes can generate it as well), but SA selection is required to generate an excess of negative values.

## Forms of sex-differential selection: Empirical data

As with previous $\hat{F}_{ST}$ metrics, we calculated unfolded reproductive $\hat{F}_{ST}$ (Eq [4]) and contrasted it against its theoretical and empirical null distributions—the latter generated by a single random permutation of LRS among the individuals of each sex. Doing so revealed that both SC and SA sites contribute to the polygenic signal of sex-differential selection affecting LRS. As

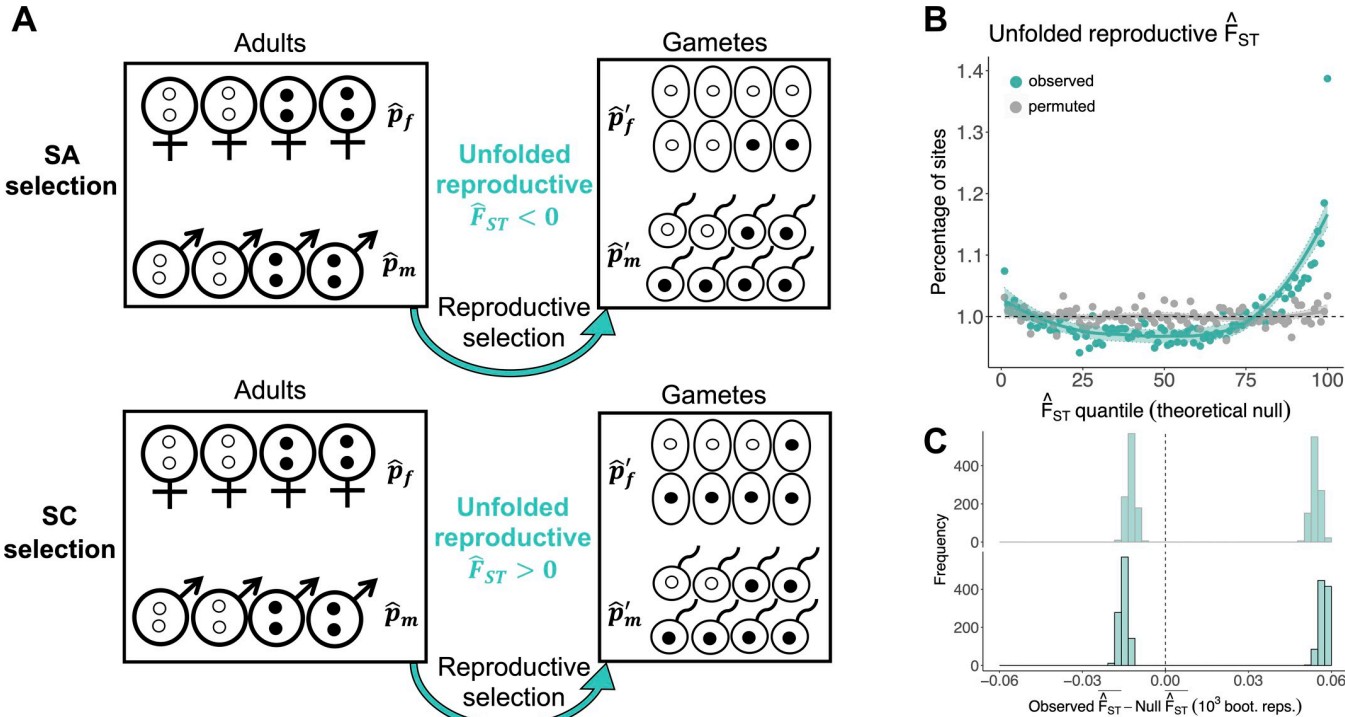

**Fig 3. Partitioning signals of sex-differential selection into SA and SC components reveals their joint contributions.** (**A**) As in Fig 1, $\hat{p}_f$, $\hat{p}_m$, $\hat{p}'_f$, and $\hat{p}'_m$ depict sex-specific frequency estimates for a given allele at different stages of the life cycle. Under SA selection (top), the white allele is female-beneficial and the black allele is male-beneficial, which tends to generate negative values of unfolded reproductive $\hat{F}_{ST}$. Under SC selection (bottom), the black allele is beneficial in both sexes, which tends to generate positive values of unfolded reproductive $\hat{F}_{ST}$. (**B**) Percentage of sites (turquoise: observed; grey: permuted) falling into each of 100 quantiles of the theoretical null distributions of unfolded reproductive $\hat{F}_{ST}$. Theoretical null data (x-axes) were generated by simulating values (nSNPs = 1,051,949) from the null (i.e., the product of 2 standard normal distributions). In the absence of sex-differential selection, approximately 1% of observed SNPs should fall into each quantile of the null (dashed line). LOESS curves (±SE) are presented for visual emphasis. (**C**) Difference, for unfolded reproductive $\hat{F}_{ST}$, between the mean observed and empirical null data (top) and between observed and theoretical null data (bottom), across 1,000 bootstrap replicates. The vertical line intersects zero, indicating no difference between the observed and null data. Differences between observed and null data were obtained separately for negative and positive values of unfolded reproductive $\hat{F}_{ST}$. This illustrates that there is enrichment of SNPs in both tails of the null. The code and data needed to generate this figure can be found at https://github.com/filipluca/polygenic_SA_selection_in_the_UK_biobank and https://zenodo.org/record/6824671. SA, sexually antagonistic; SC, sexually concordant; SNP, single-nucleotide polymorphism.

predicted under SC selection, we observed an enrichment of sites in the upper quantiles of the null distributions of unfolded reproductive $\hat{F}_{ST}$ (mean $\hat{F}_{ST}$ among sites with $\hat{F}_{ST} > 0$; theoretical null: 0.637; permuted null: 0.640; observed: 0.694; Wilcoxon and Kolmogorov–Smirnov tests, $p < 0.001$; Fig 3B and 3C). As predicted under SA selection, we observed a smaller but significant enrichment of sites in the lower quantiles of the null (mean $\hat{F}_{ST}$ among sites with $\hat{F}_{ST} < 0$; theoretical null: –0.635; permuted null: –0.638; observed: –0.651; Wilcoxon and Kolmogorov–Smirnov tests, $p < 0.001$; Fig 3B and 3C).

## Controlling for sex-specific population structure

In principle, polygenic $\hat{F}_{ST}$ elevations can arise entirely in the absence of genuine sex differences in selection if there are systematic differences in ancestry (population structure) between sexes in the sampled population [32,45]. We therefore replicated our analyses using mixed-model association tests that are analogous to $\hat{F}_{ST}$ but which explicitly correct for sex-specific population structure (see also Section F in S1 Appendix).

We first re-evaluated signals of sex differences in viability selection present in adult $\hat{F}_{ST}$ by performing a GWAS of sex [32,43,44] using standardised estimates of the log-odds ratio ($\hat{\mathcal{L}}_{ST}$; see Materials and methods). Like adult $\hat{F}_{ST}$, $\hat{\mathcal{L}}_{ST}$ quantifies between-sex allele frequency differences among adults; moreover, it controls for population structure by including a kinship matrix of genome-wide relatedness between individuals and principal components that capture structure-induced axes of genetic variation (see Materials and methods). As expected, $\hat{\mathcal{L}}_{ST}$ was highly correlated with adult $\hat{F}_{ST}$ ($r_g \pm$ SE = 1.046 ± 0.020; $p < 0.001$), and mean $\hat{\mathcal{L}}_{ST}$ was elevated relative to its empirical null distribution (null $\hat{\mathcal{L}}_{ST}$: $5.236 \times 10^{-7}$; observed: $5.323 \times 10^{-7}$; Wilcoxon and Kolmogorov–Smirnov tests, $p < 0.001$; Fig 4A and 4D), with 8.9% excess of SNPs in the top percentile of the empirical null ($\chi^2$ test, $p < 0.001$).

We then re-evaluated signals of sex-differential selection through reproductive success by performing separate GWAS for LRS in females and males, each corrected for population structure, and quantifying the difference between female and male effect sizes using a $t$-statistic ($|t|$; see Materials and methods). As expected, $|t|$ was highly correlated with reproductive $\hat{F}_{ST}$ ($r_g \pm$ SE = 1.025 ± 0.059, $p < 0.001$) and mean $|t|$ was elevated relative to its empirical null (null = 0.796, observed = 0.811, Wilcoxon and Kolmogorov–Smirnov tests, $p < 0.001$; Fig 4B and 4E), with an 11.9% excess of SNPs in the top percentile of the empirical null ($\chi^2$ test, $p < 0.001$).

We also developed an analogue of unfolded reproductive $\hat{F}_{ST}$, termed unfolded $t$ (see Materials and methods), to partition signals of sex-differential reproductive selection into SA and

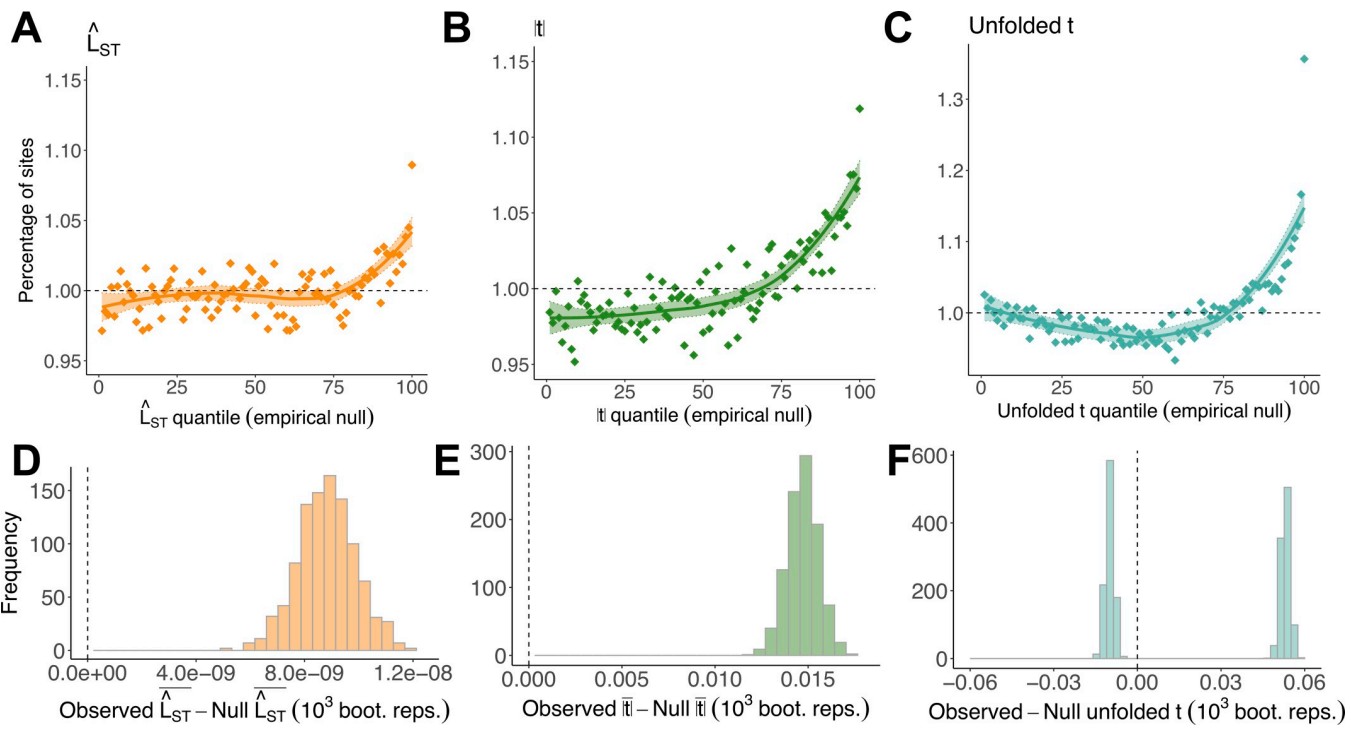

**Fig 4.** Structure-corrected metrics reaffirm $\hat{F}_{ST}$-based signals of sex-differential selection. (A–C) Percentage of sites falling into each of 100 quantiles of the empirical null distributions of $\hat{\mathcal{L}}_{ST}$, $|t|$, and unfolded $t$. In the absence of sex differences in selection, approximately 1% of observed SNPs should fall into each quantile of the null (dashed line). LOESS curves (±SE) are presented for visual emphasis. (D–F) Difference between the mean of each metric in observed and empirical null data across 1,000 bootstrap replicates. Vertical line intersects zero (no difference between observed and null data). For unfolded $t$, differences between observed and null data were obtained separately for negative and positive values. This illustrates that there is enrichment of SNPs in both tails of the null. The code and data needed to generate this figure can be found at https://github.com/filipluca/polygenic_SA_selection_in_the_UK_biobank and https://zenodo.org/record/6824671. SNP, single-nucleotide polymorphism.

SC components. As with unfolded reproductive $\hat{F}_{ST}$, SC selection should generate an enrichment of values in the upper quantiles of its null, while SA selection should generate an enrichment of values in its lower quantiles; unlike unfolded reproductive $\hat{F}_{ST}$, this metric also controls for population structure. Corroborating previous results, we observed an excess of high values of unfolded $t$ (mean $t$ among sites with $t > 0$; permuted null = 0.639, observed = 0.692, Wilcoxon and Kolmogorov–Smirnov tests, $p < 0.001$; Fig 4C and 4F) and an excess of low values of unfolded $t$ (mean $t$ among sites with $t < 0$; permuted null = –0.639, observed = –0.649, Wilcoxon and Kolmogorov–Smirnov tests, $p < 0.001$), signalling the presence of SC and SA polymorphisms, respectively.

Finally, we examined genetic correlations between metrics. These analyses showed that metrics of sex-differential LRS selection were not significantly correlated with metrics of sex-differential mortality selection across loci (Fig 5A). For example, the genetic correlation (estimated via LD score regression) between adult and reproductive $\hat{F}_{ST}$ was –0.24 (SE = 0.16, $p = 0.13$) and the genetic correlation between $\hat{\mathcal{L}}_{ST}$ and $|t|$ was –0.16 (SE = 0.16, $p = 0.31$).

## Functional and phenotypic effects of sex-differentiated loci

If sex-differentiated loci reflect genuine sex-differential selection—rather than random chance, genotyping errors, or population structure—such polymorphisms should be preferentially found in functionally important regions in the genome. We therefore conducted enrichment tests, both to support our inference that sex-differential selection is occurring and to explore functional effects of sex-differentiated loci.

We first used LD score regression [57] to test whether sites with high sex-differentiation tend to be found in major functional categories in the genome (coding, 3′UTR and 5′UTR regions). If a given category is enriched for genuine selected SNPs, the expected heritability tagged by these SNPs (i.e., what LD score regression measures) should exceed the fraction of SNPs present in that functional category. While functional enrichment estimates were noisy and thus not statistically distinguishable from 1 (no enrichment) after multiple-testing correction (Fig 5B), each estimate consistently exceeded 1 across functional categories and metrics, suggesting that sex-differentiated loci are more likely to have phenotype-altering effects than expected by chance.

Further evidence for the phenotype-altering effects of sex-differentiated loci was sought through direct comparisons between metrics of sex-differential selection and the Neale laboratory database of UK Biobank GWAS. Specifically, we used cross-trait LD score regression [58] to estimate genetic correlations between metrics of sex-differential selection and 30 phenotypes, chosen for their medical relevance and/or relationship to phenotypic sex differences. Though many significant associations did not survive multiple testing correction (Fig 5C), several disease-relevant and quantitative traits (age at menarche, body fat percentage, diseases of the eye and adnexa, fluid intelligence, injury, neuroticism score, SHBG [sex hormone binding globulin], standing height) represent candidates for sex-differential viability and LRS selection, while other traits (testosterone, high blood pressure) represent candidates for sex-differential viability selection.

## Modes of evolution of sex-differentiated loci: Theoretical predictions

To gain insight into the modes of evolution affecting sex-differentiated sites, we investigated the association between metrics of sex-differential selection and MAF in the UK Biobank. In the absence of any contemporary sex differences in selection, all between-sex $\hat{F}_{ST}$ metrics should be independent of MAF (Section G in S1 Appendix). In the presence of sex-differential

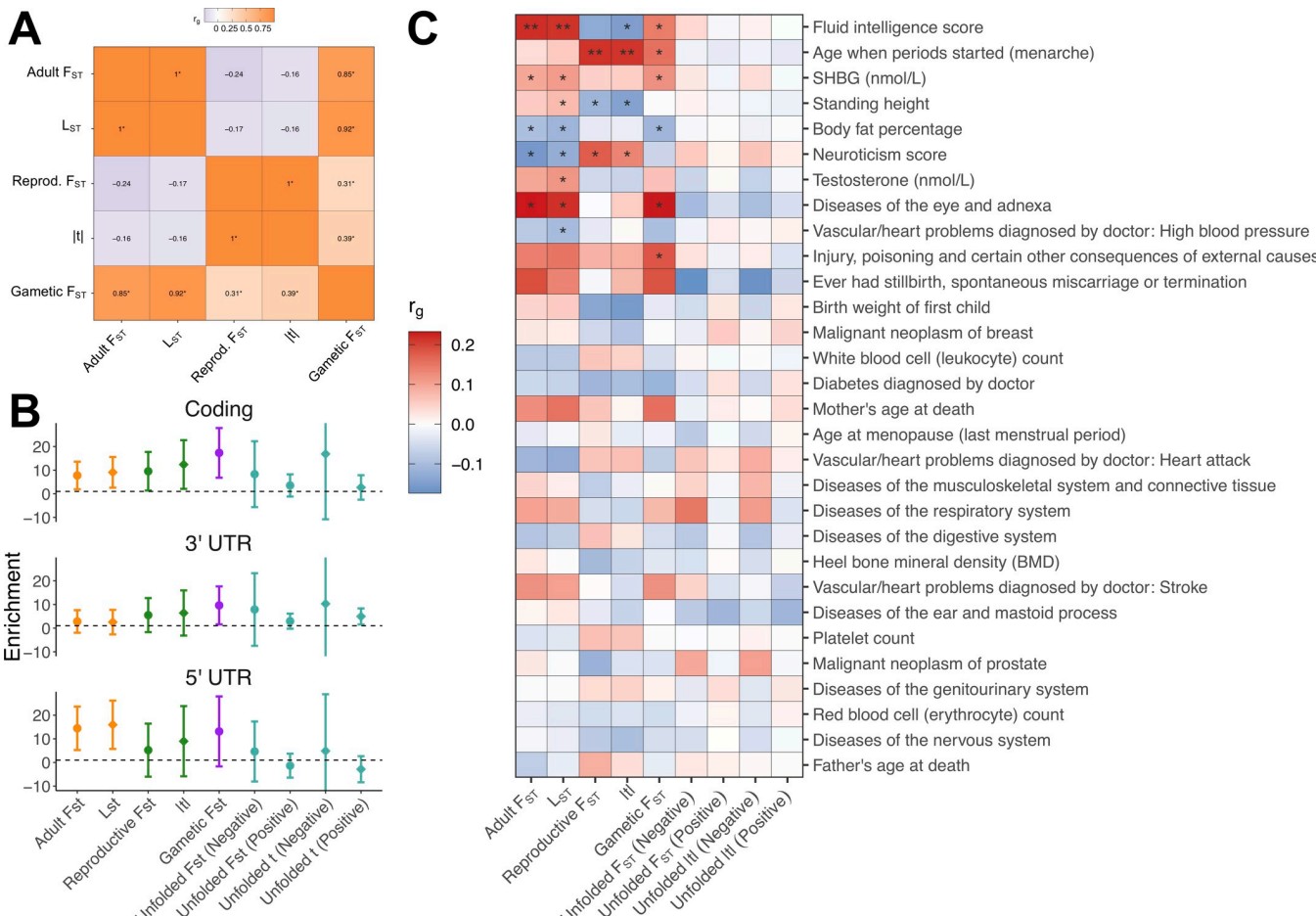

**Fig 5. Indications that sex-differentiated loci are more likely to be functional and contribute to trait variation.** (**A**) Genetic correlations between metrics of sex-differential selection. Positive correlations (orange) imply that alleles have similar sex-specific effects on given fitness components, while negative correlations (purple) imply that alleles have opposing sex-specific effects on given fitness components; * denotes unadjusted $p < 0.05$. (**B**) Enrichments (±SE) of sex-differentiated loci in major functional categories. For each metric, enrichments were calculated as the relative SNP heritability (as a fraction of total SNP heritability) explained by a given functional category, divided by the relative number of SNPs (as a fraction of all SNPs) present in a given functional category. Dashed line = 1 (no enrichment). "Negative" and "Positive" refer to negative and positive values (i.e., SA and SC components, respectively) of unfolded reproductive $\hat{F}_{ST}$ and unfolded $t$ metrics. (**C**) Genetic correlations between metrics of sex-differential selection and various UK Biobank phenotypes (as analysed by the Neale laboratory). Metrics of sex-differential selection have been polarised, such that positive correlations (red) suggest that higher trait values are more beneficial to females than males (for the relevant fitness component), while negative correlations (blue) suggest that higher trait values are more beneficial to males than females (see Discussion for caveats surrounding this interpretation); ** denotes FDR-adjusted $p < 0.05$ and * denotes unadjusted $p < 0.05$. The code needed to generate this figure can be found at https://github.com/filipluca/polygenic_SA_selection_in_the_UK_biobank and https://github.com/lukeholman/UKBB_LDSC, with data at https://zenodo.org/record/6824671. FDR, false discovery rate; SA, sexually antagonistic; SC, sexually concordant; SNP, single-nucleotide polymorphism.

selection, the association between each metric and MAF can potentially be positive or negative, depending on the patterns of contemporary and historical selection affecting loci throughout the genome. A positive covariance between $\hat{F}_{ST}$ and MAF should arise when alleles subject to sex-differential selection often segregate at intermediate frequencies, as may occur under a history of balancing selection or drift (Section G in S1 Appendix) or non-equilibrium scenarios such as incomplete selective sweeps. In contrast, a negative association between MAF and between-sex $\hat{F}_{ST}$ is expected for loci that have evolved under sex-differential purifying selection (Section G in S1 Appendix). This negative covariance arises because purifying selection disproportionately lowers the frequency of large-effect alleles (those generating larger $\hat{F}_{ST}$ values)

relative to small-effect alleles [59]. In short, positive associations with MAF indicate that purifying selection is not the dominant mode of evolution affecting loci under sex-differential selection and instead signal a recent history of balancing selection, positive selection, or drift.

While associations between metrics of sex-differential selection and MAF provide insights into relatively recent and contemporary patterns of selection affecting sex-differentiated sites, they do not provide insights into their deeper evolutionary histories. To examine this, we tested the specific hypothesis that sex-differentiated sites are subject to long-term balancing selection, as predicted for SA polymorphisms under certain scenarios of selection and dominance [10]. Under long-term balancing selection, we would expect sex-differentiated (and linked) loci to be old, to exhibit low between-population $\hat{F}_{ST}$, to exhibit high genetic diversity, and to disproportionately co-localise with previous candidates for long-term balancing selection, compared to less sex-differentiated sites with similar allele frequencies in the UK Biobank.

### Modes of evolution of sex-differentiated loci: Empirical data

Examining the relationship between MAF and metrics of sex-differential selection in the UK Biobank data revealed consistently positive correlations (adult $\hat{F}_{ST}$, $\rho = 0.009$, $p < 0.001$; $\hat{\mathcal{L}}_{ST}$: $\rho = 0.006$, $p = 0.216$; reproductive $\hat{F}_{ST}$, $\rho = 0.006$, $p < 0.001$; $|t|$: $\rho = 0.005$, $p < 0.001$; gametic $\hat{F}_{ST}$, $\rho = 0.007$, $p < 0.001$; Fig 6A–6D), with all correlations stronger in observed than null data (Section H in S1 Appendix). Given the absence of negative correlations between MAF and each metric, we can reject purifying selection as the dominant mode of evolution affecting sex-differentiated sites. The positive correlations instead suggest that balancing selection, drift, or incomplete selective sweeps characterise the evolution of sex-differentiated loci.

We then tested the hypothesis that long-term balancing selection has shaped the evolutionary histories of sex-differentiated loci. We focused our analyses on 4 measures of balancing selection: allele age estimates from the Atlas of Variant Age database [60], between-population $\hat{F}_{ST}$ and Tajima's D estimates from 2 non-European populations from the 1000 Genomes Project [61], and 3 sets of candidate loci for long-term balancing selection [62–64]. In each case, we looked for associations between metrics of sex-differential selection and balancing selection, while controlling for ascertainment bias of intermediate-frequency alleles (which are, on average, older and thus more likely to be under long-term balancing selection irrespective of the strength of sex-differential selection) among highly sex-differentiated sites (see Materials and methods). Overall, we found little support for the hypothesis of long-term balancing selection affecting sex-differentiated loci. After corrections for multiple testing across metrics of sex-differential selection (see Section I in S1 Appendix, for full statistical results), we found weak or absent associations with allele age (Fig 6E–6H), between-population $\hat{F}_{ST}$ (Section I in S1 Appendix), genetic diversity (Section I in S1 Appendix), or previous candidates for balancing selection (Section I in S1 Appendix). We found some indications that candidate SA alleles (i.e., loci with negative values of unfolded reproductive $\hat{F}_{ST}$ and unfolded $t$) were older than the genome-wide average (Fig 6H), and loci experiencing strong SC selection (i.e., positive values of unfolded reproductive $\hat{F}_{ST}$ and unfolded $t$) were younger (Fig 6H).

## Discussion

Sex differences in directional selection on phenotypes have been reported in a wide range of animal taxa [19,21–23,65], including post-industrial human populations [28–30], yet population genomic signals of sex-differential selection—let alone SA selection—have been extremely difficult to establish. The reason is simple: Sexual reproduction equalises autosomal allele

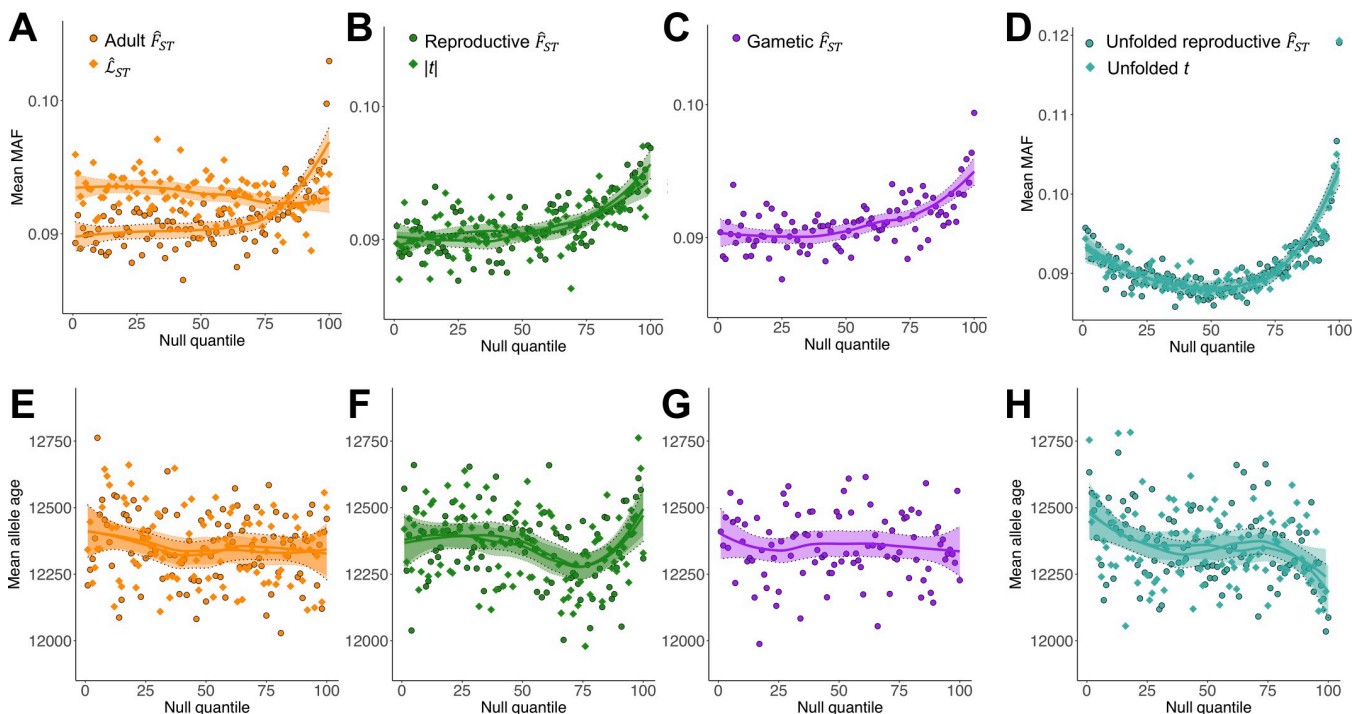

**Fig 6. Modes of evolution of sex-differentiated sites. (A–D)** Mean MAF, in the UK Biobank, across 100 quantiles of the null for each metric of sex-differential selection. For $\hat{F}_{ST}$ metrics, x-axes correspond to Fig 2A–2C (and Fig 3B for unfolded reproductive $\hat{F}_{ST}$). For mixed-model metrics, x-axes correspond to Fig 4A–4C. LOESS curves (±SE) are presented for visual emphasis. **(E-H)** Mean age of the alternative (i.e., non-reference) allele across 100 quantiles of the null for each metric of sex-differential selection. Each panel corrects for ascertainment bias of allele frequencies among highly sex-differentiated sites (i.e., Fig 6A–6D). For visualisation purposes, this was done by averaging, in each quantile, allele age across 20 quantiles of alternative allele frequency in the UK Biobank (such that UK Biobank alternative allele frequency is approximately equal across quantiles). LOESS curves (±SE) are presented for visual emphasis. The code and data needed to generate this figure can be found at https://github.com/filipluca/polygenic_SA_selection_in_the_UK_biobank and https://zenodo.org/record/6824671.

frequencies between the sexes every generation, restricting genetic divergence and, in effect, preventing the use of common tests to infer sex differences in selection (e.g., McDonald–Kreitman tests for positive selection, $F_{ST}$ outlier tests for spatially varying selection [66–68]). Published studies using human genomic data illustrate the challenges of studying polymorphisms with sex-differential fitness effects [32,45], including sample sizes that may be insufficient for detecting polygenic signals of sex-differential selection, lack of controls for population structure or technical artefacts, and/or absence of data concerning reproductive fitness components.

## Signals of sex-differential selection in the UK Biobank

We developed a theoretical framework for studying genomic variation with sex-differential effects across a complete life cycle. Our approach extends current work based on between-sex allele frequency differentiation among adults—a potential signal of sex-differential viability selection among juveniles [32,34,45]—to further include reproductive success components and total fitness. Applying this approach to data from a quarter-million UK adults, we present evidence for polygenic signals of sex-differential selection in humans. Specifically, UK Biobank individuals showed sex differences in allele frequencies—both among adults and their (projected) offspring—that consistently exceeded expectations defined by our theoretical null models for viability, reproductive, and total fitness and persisted after controlling for potential artefacts arising from mis-mapping of reads to sex chromosomes [44].

Although we focussed on $F_{ST}$ as our metric of differentiation for a variety of reasons (its simplicity, amenability to theoretical modelling, and rich history in population genetic studies of adaptation [66–68]), an important drawback of $F_{ST}$ is its inability to control for systematic sex differences in the genetic ancestry of sampled individuals. We therefore used $F_{ST}$ analogues based on mixed-model association tests to control for sex-specific population structure. These $F_{ST}$ analogues corroborated $F_{ST}$-derived signals of sex-differential selection on each component, with clear enrichments in the upper tails of each null distribution. Additional support for genuine sex-differential selection came from functional enrichment analyses, which, despite noisy individual estimates, consistently indicated that sex-differentiated sites were situated in functional genomic regions and contributed to variation for many phenotypes.

An important limitation of metrics of sex-differential selection affecting non-LRS fitness components (i.e., adult $F_{ST}$, gametic $F_{ST}$, and their mixed-model analogues) applied to the UK Biobank is that UK Biobank individuals are sampled through active participation. Consequently, as noted by Pirastu and colleagues [43], sex differences in the genetic basis of individuals' predisposition to take part in the UK Biobank may generate sex differences in adult allele frequencies. To support this argument, Pirastu and colleagues [43] reported significantly greater SNP heritability of sex (a polygenic measure of sex differences in allele frequencies) in biobanks relying on active participation than in biobanks using passive participation. However, their analysis is inconclusive because the passive participation studies they analysed were smaller ($N_{Biobank\ Japan}$ = 178,242, $N_{FinnGen}$ = 150,831, $N_{iPsych}$ = 65,891) than active participation studies ($N_{UK\ Biobank}$ = 452,302, $N_{23andme}$ = 2,462,132). Thus, differences in statistical power between studies (and/or differences in the extent of sex-differential viability selection between populations) could account for their results. Moreover, the positive point estimates of SNP heritability for passive participation studies suggest that substantial allele frequency differences between the sexes are possible. For example, mortality after fertilisation, but before birth, is very high in humans (on the order of 50% [69]), giving ample opportunity for mortality in early life to generate allele frequency differences between sexes. In sum, neither their study nor ours can conclusively distinguish the relative contributions of sex-differential selection and participation bias to allele frequency differentiation between female and male adults, though both sources likely contribute.

Importantly, participation bias should not affect metrics of sex-differential selection relating to LRS. Reproductive $\hat{F}_{ST}$ and its mixed-model analogue, $|t|$, control for allele frequency differences between samples of adults of each sex and rule out factors that might otherwise affect estimated adult allele frequencies in the UK Biobank (e.g., mis-mapping of reads to sex chromosomes, participation biases [43]). Elevations in these metrics thus provide the most compelling evidence for sex-differential selection in the UK Biobank (see also [46]). Moreover, they are consistent with previous observations in post-industrial human populations, including variation in female and male LRS [70] (a necessary precondition for sex-differential selection), widespread sex differences in the genetic basis of quantitative traits (e.g., in the UK Biobank [71]), and sex-differential selection on phenotypes (e.g., height [29,30] and multivariate trait combinations [70]), which should collectively lead to genome-wide polymorphisms with sex-differential effects on fitness and fitness components [20].

## Distinguishing between SA and SC forms of sex-differential selection

Having established signals of sex-differential selection affecting LRS, we developed a new test for investigating the form of selection—SC or SA—affecting these genomic variants by quantifying the product of a genetic variant's effect on LRS in each sex. Applying our test to UK Biobank data showed that both types of variant contribute to signals of sex-differential selection

on LRS, with SC variants contributing comparatively more enrichment in the upper tail of the null of unfolded reproductive $\hat{F}_{ST}$ (and its mixed-model analogue, unfolded $t$) than SA variants contribute in the lower tail of the null. That signals of SC polymorphism were more pronounced than SA polymorphism is perhaps unsurprising, given that most traits are likely to be subject to SC rather than SA selection [29]. Moreover, alleles subject to identical SC selection in each sex will contribute to the upper tail of unfolded reproductive $\hat{F}_{ST}$, but will not contribute to the lower tail (or to other metrics of sex-differential selection), which might also account for greater apparent signal of SC than SA selection in these analyses. Nonetheless, some human traits have been shown to be under SA selection—most notably standing height, which positively covaries with male LRS and negatively covaries with female LRS [28–30]. The enrichment of sites in the lower tails of unfolded reproductive $\hat{F}_{ST}$ and unfolded $t$ is consistent with these previous observations. Our finding that variants that increase height tended to have male-beneficial and female-detrimental effects (i.e., as reflected by a negative correlation between height and $t$) is particularly reassuring and validates the intuition that SA selection at the phenotypic level (e.g., over height) gives rise to SA variation throughout the genome.

## Modes of evolution affecting sex-differentiated loci

We found that sex-differentiated sites had, on average, more intermediate frequencies than less sex-differentiated sites. This finding has several implications. First, we expect no association between metrics of sex-differentiation and MAF in the absence of sex-differential selection. Therefore, these positive associations represent an independent strand of support for the argument that sex-differential selection is shaping patterns of genome-wide variation in the UK Biobank. Second, the positive associations imply that a model of sex-differential purifying selection, in which variants are maintained at mutation-selection-drift balance, is inadequate to explain enrichments of sex-differentiated sites. Sex-differential purifying selection is instead expected to generate negative associations between MAF and the extent of sex-differentiation (a negative association that is indeed observed for many quantitative traits [72]). Finally, the positive associations between sex-differentiation and MAF are consistent with a variety of scenarios, such as recent evolutionary histories of balancing selection, genetic drift, or incomplete selective sweeps. Balancing selection or drift can both generate a broad spectrum of allele frequency states at SA loci, in which intermediate-frequency SA variants dominate signals of sex-differential selection. Alternatively, SC alleles with unequal fitness effects in each sex could have recently swept to intermediate frequencies and these variants now dominate signals of sex-differential selection.

Although positive associations between metrics of sex-differential selection and MAF indicate that balancing selection may be present, our analyses did not reveal clear signals of long-term balancing selection among sex-differentiated sites. The absence of such signals may stem from several factors. First, SA polymorphisms are only predicted to experience balancing selection under narrow conditions [10,73], so SA loci may not experience balancing selection at all. Second, balancing selection could affect sex-differentiated polymorphisms but be too recent to generate a clear statistical signal in our analyses [74]. Third, long-term balancing selection at sex-differentiated loci may be present but effectively weak, owing to relatively small $N_e$ in humans [75] and the high susceptibility of SA alleles to genetic drift [73,76]. Fourth, long-term balancing selection may be present, but statistical tests for it may be too weak to stand out from the background noise of false positives in our metrics and the datasets used to quantify balancing selection [77].

How do we reconcile these results with previous work in *Drosophila melanogaster* indicating that candidate SA polymorphisms segregate across worldwide populations and even

species [33]? A parsimonious explanation for these contrasting findings is that the effectiveness of balancing selection is lower in humans than fruit flies due to much smaller $N_e$. Indeed, given the pronounced sensitivity of SA balancing selection to genetic drift [73,76], we should expect the relationship between signals of SA and balancing selection to vary with $N_e$. Moreover, previous work in *D. melanogaster* focussed on SA polymorphisms [33] to the exclusion of SC polymorphisms, whereas our metrics capture both forms of sex-differential variation, thus weakening the power of tests for associations with signals of balancing selection. Interestingly, when we partitioned signals of sex-differentiation into SA and SC components, we found indications that candidate SA sites were indeed older, which implies that SA balancing selection may be present but masked by sex-differential SC polymorphisms. Overall, evidence that sex-differentiated, including SA, polymorphisms contribute to standing genetic variation —as in our study—is at present much stronger than evidence that they are maintained by balancing selection.

## Directions for future research

Our analyses suggest a number of fruitful directions for further research. First, given the difficulty of distinguishing participation bias from selection in signals of between-sex allele frequency differentiation among adults, conclusively establishing the presence of sex-differential viability selection in genomic data remains an important research direction. Parent-offspring trio analyses that control for participation effects [78], or replication of our analysis strategy in large datasets sampled through passive rather than active participation, may yield the evidence required. Second, the extent to which variants with positive effects on mortality in a given sex have similar or opposing effects on reproduction bears further examination. Our finding that genetic correlations between metrics of viability and reproductive selection were not significantly different from zero indicates a range of possible scenarios. It may suggest that variants affecting each fitness component are independent (i.e., because alleles affecting each component are genuinely independent), that between-sex allele frequency differentiation among adults is a poor signal of sex-differential viability selection or that a similar fraction of loci have concordant and antagonistic effects, thus also generating no net correlation.

Finally, given the increasing availability of genotypic and LRS data, further work could attempt to replicate our analysis strategy in different populations and species. Many taxa exhibit greater variance for reproductive success than humans [79], generating higher potential for detecting polygenic signals of sex-differential selection. In line with this, polygenic inflations of adult $\hat{F}_{ST}$ have previously been documented in modest samples of pipefish and flycatchers [32,38,39], suggesting that sex differences in selection might be stronger in those species than in humans. Moreover, these samples are less susceptible to ascertainment bias because individuals do not actively participate and because sampling can often be randomised with respect to sex. While we expect that polygenic signals of sex-differential selection will replicate across populations of a species (see, for example, Zhu and colleagues [35]'s replication of the association between testosterone and adult allele frequency differences in Fig 5C), we caution that there may be relatively little overlap in terms of the most sex-differentiated polymorphisms. One reason is that environmental differences between populations (e.g., cultural differences in family planning between human populations) could alter the set of causal loci under sex-differential selection. Another reason is that the noisiness of polygenic signals of sex-differential selection [32,45], along with the near certainty that most polymorphic loci have small effects on fitness [80], generates variation in the set of candidate sex-differential polymorphisms identified across populations [81], even if causal sex-differential polymorphisms do not differ.

## Materials and methods

### Ethics statement

The UK Biobank has Research Tissue Bank approval from the North-West Multi-centre Research Ethics Committee. Approval for using UK Biobank data for this specific project, from participants consenting to share anonymised data, was granted under project number 52049.

### Quality control of UK Biobank data

We used sample-level information provided by the UK Biobank (see [55] for details) to perform individual-level (phenotypic) quality controls. Specifically, we excluded individuals with high relatedness (third degree or closer), non-"white British" ancestry, high heterozygosity, and high missing rates. We also excluded individuals whose reported sex did not match their inferred genetic sex, aneuploids, and individuals with missing or unreliable LRS data (as detailed below).

We processed LRS data as follows. LRS data were obtained from UK Biobank field 2405 "Number of children fathered" for males, and field 2734 "Number of live births" for females. Previous observations of positive genetic correlations between offspring and grand-offspring numbers across generations [82] indicate that offspring number represents a good proxy for LRS in post-industrial human populations. Because some individuals were asked to report offspring number at repeated assessment points, we considered the maximum offspring number reported as the definitive value of LRS for that individual. Though misestimation of LRS for each individual cannot be definitively excluded (e.g., individuals may misreport and include non-biological children, individuals may reproduce after data collection), we minimised this possibility by removing individuals: (i) younger than 45 years of age (this cutoff was chosen for consistency with previous research [29] and because Office for National Statistics data indicates that reproduction is very limited for UK individuals aged 45 and over); (ii) reporting fewer offspring at a later assessment point than at an earlier assessment point; (iii) with 20 or more reported offspring numbers (large offspring numbers often ended in zero—e.g., 20, 30, 50, 100—and were thus considered less reliable). Furthermore, uncounted LRS data add imprecision but should not systematically bias our analyses.

In addition to site-level quality controls implemented by the UK Biobank [55], we used PLINK and PLINK2 [83] to remove imputed sites that were non-diallelic, had MAF <1%, missing rates >5%, $p$-values $< 10^{-6}$ in tests of Hardy–Weinberg equilibrium, and INFO score ≤0.8, denoting poor imputation quality. While these cutoffs restrict our analyses to a nonrandom subset of all genetic variation, they guard against sequencing artefacts in the UK Biobank and help remove sites (e.g., those with MAF <1%) which have little potential to carry statistical signal of sex-differentiation relative to noise induced by sampling error.

### Additional artefact filtering in UK Biobank data

Mis-mapping of autosomal reads to sex chromosomes can generate between-sex allele frequency differences among adults in the absence of sex differences in selection [44]. In light of scant direct evidence for SA polymorphisms in humans and still-developing bioinformatic methods for distinguishing artefacts from genuine sex-differential selection [40,44,84–86], our primary concern was to reduce the chance of mapping errors. We did so by excluding: (i) sites with heterozygosity levels that exceeded what could plausibly be expected under SA selection (see below and Section C in S1 Appendix); (ii) sites with a deficit of minor allele homozygotes; and (iii) sites exhibiting large differences in missing rate between sexes. These 3 patterns have

previously been shown to correlate with mis-mapping of reads to sex chromosomes [44]. While these filters reduce the chance of false positives, they also potentially increase chance of false negatives and therefore represent a slightly conservative test of sex-differential selection. For example, the removal of sites with high heterozygosity levels is expected to remove sites under strong (but not weak or moderately strong) sex-differential selection; similarly, the removal of sites with large missing rate differences between sexes may remove genuine poly-morphisms with sex-differential effects.

To remove sites with artificially inflated heterozygosity, we estimated $F_{IS}$ for each SNP as:

$$\hat{F}_{IS} = \frac{P_{Aa}}{2\bar{p}(1-\bar{p})} - 1$$

where $P_{Aa}$ denotes the frequency of heterozygotes for a given locus and $\bar{p}$ the sex-averaged allele frequency. For a SA locus at polymorphic equilibrium, the distribution of $\hat{F}_{IS}$ is well approximated by a normal distribution with expectation and variance as follows:

$$\text{E}[\hat{F}_{IS}] \approx \frac{1}{2n} + \frac{p(1-p)}{4}\left(\frac{s_{\max}}{1-ps_{\max}}\right)^2$$

$$\text{var}[\hat{F}_{IS}] = \frac{1}{n}$$

where $n$ is total sample size of adults, $p$ the minor allele frequency, and $s_{\max} = \max(s_m, s_f)$ with $s_m$ and $s_f$ representing male and female selection coefficients (Section C in S1 Appendix). To identify SNPs with excess heterozygosity, we compared $\hat{F}_{IS}$ in the observed data to expected $\hat{F}_{IS}$ under strong SA selection ($s_{\max} = 0.2$) by performing a 1-tailed Z-test for excess heterozygosity. We thus obtained $p$-values for each locus, corrected $p$-values for multiple testing using Benjamini–Hoch-berg false discovery rates (FDR) [87], and removed sites with FDR q-values below 0.05.

To identify sites with a deficit of minor allele homozygotes, we compared the observed fre-quency of minor allele homozygotes to the expected frequency under Hardy–Weinberg equi-librium ($p^2$, where $p$ is the frequency of the minor allele) by performing a 1-tailed binomial test, removing sites with FDR q-values below 0.05. Tests for excess heterozygosity and deficits of minor allele homozygotes were performed across all individuals (regardless of sex) and also for each sex separately. Sites were removed if they exhibited q-values below 0.05 in any of the 3 tests (i.e., both sexes combined, females, and males). Finally, to assess differences in missing rate between the sexes, we performed a $\chi^2$ test, removing sites with FDR q-values below 0.05.

## Quantifying polygenic signals of sex differences in selection

$\hat{F}_{ST}$-**based metrics.** We used $\hat{F}_{ST}$ to quantify allele frequency differences between sexes. $\hat{F}_{ST}$ is a simple metric, well established in evolutionary biology research, amenable to theoreti-cal modelling (as in Eqs [3A–3C]), and independent of MAF in the absence of sex-differential selection (unlike, say, raw allele frequency differences [32]). We obtained allele frequencies in adults of each sex directly from sequence data (after filtering individuals and sites, as described above) and used them to calculate adult $\hat{F}_{ST}$ for each polymorphic site. We obtained allele fre-quencies among projected gametes using LRS data (as per Eq [1]) and used them to calculate reproductive $\hat{F}_{ST}$, gametic $\hat{F}_{ST}$, and unfolded reproductive $\hat{F}_{ST}$ (as per Eqs [2A–2C] and [4]).

**Statistical comparisons of null and observed distributions.** Null distributions for $\hat{F}_{ST}$ metrics were theoretically derived (see Sections A and E in S1 Appendix). The theoretical null distributions apply to genome-wide data in which the sample of female and male sequences,

mean and variance in LRS, and Hardy–Weinberg deviations, are constant across loci. In practice, there is variation in sample sizes, mean LRS, variance in LRS, and the extent of Hardy–Weinberg deviations between loci. To take these factors into account, we let the multiplier in Eqs [3A–3C] vary in terms of its sample size ($N_{f_i}$ and $N_{m_i}$ per diploid locus $i$), mean and variance in LRS ($\mu_{f_i}$ and $\mu_{mi}$, and $\sigma_f^2 i$ and $\sigma_m^2 i$, per diploid locus $i$) and the extent of Hardy–Weinberg deviations in the sample ($\hat{F}_{IS}^f i$ and $\hat{F}_{IS}^m i$ per diploid locus $i$). We then scaled $\hat{F}_{ST}$ by the multiplier, such that, for each locus:

$$\hat{F}_{ST(Adult,scaled)} \approx \frac{\hat{F}_{ST(Adult)}}{\left(\frac{1}{8N_{f_i}} + \frac{1}{8N_{m_i}}\right)} \approx X$$

$$\hat{F}_{ST(Reprod.,scaled)} \approx \frac{4\hat{p}(1-\hat{p})\hat{F}_{ST(Reprod.)}}{\frac{\hat{p}_f(1-\hat{p}_f)}{2N_{f_i}}\frac{\sigma_f^2 i}{\mu_f^2 i}\left(1 - \hat{F}_{IS}^f i\right) + \frac{\hat{p}_m(1-\hat{p}_m)}{2N_{m_i}}\frac{\sigma_m^2 i}{\mu_m^2 i}\left(1 - \hat{F}_{IS}^m i\right)} \approx X$$

$$\hat{F}_{ST(Gametic,scaled)} \approx \frac{\hat{F}_{ST(Gametic)}}{\left[\frac{1}{8N_{f_i}}\left(1 + \frac{\sigma_f^2 i}{\mu_f^2 i}\right) + \frac{1}{8N_{m_i}}\left(1 + \frac{\sigma_m^2 i}{\mu_m^2 i}\right)\right]} \approx X.$$

These scaled $\hat{F}_{ST}$ estimates, which correct for site-specific variation, can then be compared to a chi-square distribution with 1 degree of freedom. For unfolded reproductive $\hat{F}_{ST}$, no scaling is required because site-specific adjustments are already taken into consideration in the definition of the metric (Eq [4]).

Null distributions were also obtained empirically, through permutation, as follows. For adult and gametic $\hat{F}_{ST}$, we performed a single permutation of female and male labels and recalculated $\hat{F}_{ST}$ (scaled by the multiplier, as above) in permuted data. For reproductive and unfolded reproductive $\hat{F}_{ST}$, we performed a single permutation of LRS values within each sex —without permuting sex—and recalculated the statistic (scaled by the multiplier, as above) in permuted data. Permuting LRS without permuting sex is appropriate for reproductive and unfolded reproductive $\hat{F}_{ST}$ because it allows allele frequencies to differ between adult males and females (as would happen if, for example, sex-differential viability selection is occurring among juveniles) but randomises the effects of genotype on LRS, thus ensuring that only estimation error can contribute to the empirical null. We performed a single permutation for each metric because performing large numbers of permutations was computationally unfeasible and because we were focussed on testing a cumulative signal of selection across loci, rather than establishing significance at the single-locus level.

To test for elevations in observed data relative to the (theoretical or empirical) nulls, we LD-pruned the dataset (settings "—indep-pairwise 50 10 0.2" in PLINK) and ran Wilcoxon rank-sum and Kolmogorov–Smirnov tests. These tests assess differences in the median and distribution of the observed and null data, respectively. As a complementary way of comparing observed and null data, we quantified enrichment of observed values in the top 1% of each null using a $\chi^2$ test. Finally, we estimated the difference between the mean value of the metric in the observed data and the mean value of the metric in each null, obtaining 95% confidence intervals and empirical $p$-values through bootstrapping (1,000 replicates; where each replicate consists of the set of relevant SNPs, sampled with replacement).

## Controlling for sex-specific population structure

**Case-control GWAS of sex.** To complement the test for sex-differential viability selection based on adult $\hat{F}_{ST}$, we performed a GWAS of sex [32,43,44]. By analogy to adult $\hat{F}_{ST}$, loci with sex-differential effects on viability in a GWAS of sex will tend to have relatively large absolute log-odds ratios (corresponding to relatively large allele frequency differences between sexes). Unlike adult $\hat{F}_{ST}$, the GWAS of sex approach additionally permits the inclusion of covariates that account for population structure and other possible confounders [32,43,44].

We used BOLT-LMM to run a mixed-model GWAS [88] using a kinship matrix to account for population structure. The kinship matrix was constructed from an LD-pruned set of quality-filtered imputed SNPs (LD-pruning settings as above). We added individual age (field 54), assessment centre (field 21003), and the top 20 principal components derived from the kinship matrix, as fixed-effect covariates. To facilitate comparisons with adult $\hat{F}_{ST}$, we standardised the regression coefficients (log-odds ratios) from the GWAS by allele frequency, such that:

$$\hat{\mathcal{L}}_{ST} = (\sqrt{\bar{p}(1-\bar{p})}\hat{\mathcal{L}})^2$$

where $\hat{\mathcal{L}}$ is the log-odds ratio and $\bar{p}$ is the sex-averaged allele frequency among adults. To obtain permuted $\hat{\mathcal{L}}_{ST}$ values, we performed a single permutation of female and male labels and recalculated the statistic in the permuted data.

**$t$-Statistics for sex-differential effects on LRS.** To complement the test for sex differences in selection on LRS based on reproductive $\hat{F}_{ST}$, we performed a GWAS of LRS in each sex separately, using a mixed-model GWAS, which allowed us to correct for population structure in effect size estimates. Following [89], we quantified differences in male and female effect size estimates by means of a $t$-statistic, defined as:

$$|t| = \left| \frac{\beta_{FM} - \beta_{FM}}{\sqrt{SE_{MF}^2 + SE_{FM}^2 - 2\rho SE_{MF}SE_{FM}}} \right|$$

where $\beta_F$ and $\beta_M$ are estimated effect sizes obtained from sex-stratified GWAS (implemented in BOLT-LMM as above), $SE_M$ and $SE_F$ are sex-specific standard errors, and $\rho$ is the between-sex rank correlation among genome-wide LD-pruned loci. Relative to permuted data (obtained using an identical procedure to that implemented for reproductive $\hat{F}_{ST}$), loci with elevated $|t|$ in observed data denote candidate loci with sex-differential effects on LRS.

To examine the relative contributions of SA and SC components to signals of sex-differential reproductive selection, while correcting for sex-specific population structure, we developed an analogue of unfolded reproductive $\hat{F}_{ST}$, termed "unfolded $t$," as:

$$\text{Unfolded } t = \frac{\beta_F \beta_M}{\sqrt{SE_F^2 SE_M^2}}.$$

Relative to permuted data, SA selection (i.e., opposing signs of $\beta_F$ and $\beta_M$) will tend to generate an excess of negative values of unfolded $t$, while SC selection (i.e., same sign of $\beta_M$ and $\beta_F$) will tend to generate an excess of positive values of unfolded $t$.

## Functions and phenotypic effects of sex-differentiated loci

We used stratified LD score regression [57] to examine whether sex-differentiated loci were more likely to be situated in putatively functional genomic regions (e.g., coding or UTR regions) than expected by chance. This method partitions the heritability from GWAS

summary statistics into different functional categories, while accounting for differences in LD (and thus, increased tagging of a given causal locus) in different regions of the genome (with LD quantified from European-ancestry samples from the 1000 genome project, and restricted to SNPs also present in the HapMap 3 reference panel [57]). Because LD score regression requires signed summary statistics as input, we first transformed our (unsigned) metrics of sex-differential selection to signed metrics (e.g., $\hat{F}_{ST}$ metrics and $\hat{\mathcal{L}}_{ST}$ were transformed to Z-scores, $|t|$ was transformed to $t$), where positive and negative values denote female- and male-beneficial effects of the focal allele, respectively.

Enrichments for 3 putatively functional categories (coding, 3′UTR, 5′UTR) were then calculated as the fraction of total heritability explained by a given category divided by the fraction of all SNPs in a given category. Note that we calculated enrichment for these categories while implementing the "full baseline model," which includes 50 further categories. This model has been shown to provide unbiased enrichments for focal categories [57] and for total SNP heritability [90] (estimates of total SNP heritability were used in Section D in S1 Appendix).

We used cross-trait LD score regression [58] to examine genetic correlations between metrics of sex-differential selection and a suite of phenotypic traits, as well as between the metrics of sex-differential selection. The method calculates genetic correlations between pairs of traits while taking into account LD-induced differences in the extent of tagging of causal loci across the genome. We computed genetic correlations between each metric of sex-differential selection (transformed to a signed statistic, as above, such that higher values of the signed metric are more likely to benefit females than males) and an initial list of 43 traits (subsequently filtered to 30 after removing traits where an accurate genetic correlation, defined as SE < 0.2, could not be estimated) (http://www.nealelab.is/uk-biobank/), and used FDR correction (across metrics and traits) on resulting $p$-values.

## Modes of evolution affecting sex-differentiated loci

**Associations with MAF in the UK Biobank.**   To test for associations between metrics of sex differences in selection and MAF, we estimated a Spearman's rank correlation between each metric and MAF. We also tested whether the relationship between metrics of sex differences in selection and MAF was more pronounced in the observed data than in the (theoretical or empirical) nulls by estimating the difference between correlations in the observed data and correlations in the null data among 1,000 bootstrap replicates (as above), thereby generating 95% confidence intervals and empirical $p$-values.

**Allele ages.**   If sex-differentiated variants experience sufficiently strong and sustained balancing selection relative to the countervailing effects of genetic drift, we expect them to be older than the genome-wide average [74]. We used the Atlas of Variant Age database to obtain allele age estimates for genome-wide variants [60]. Estimates of allele age in this database apply to the non-reference (i.e., alternative) allele and are derived from coalescent modelling of the time to the most recent common ancestor using the "Genealogical Estimation of Variant Age" method (see [60] for details). Estimates of allele age make use of genomic data from: (i) the 1000 Genomes Project; (ii) the Simons Genome Diversity Project; and (iii) both datasets combined. For each site in the UK Biobank, we obtained the median estimate of allele age from the combined dataset (when available), from the 1000 Genomes Project, or the Simons Genome Diversity Project (when neither alternative estimate was available).

**Between-population $F_{ST}$ and Tajima's D in non-European populations.**   If candidate SA variants experience sufficiently strong balancing selection maintaining a fixed polymorphic equilibrium, they should exhibit lower-than-average allele frequency differences between populations [74] and larger-than-average allele frequency diversity within populations. We used

bcftools [91] to obtain allele frequency data from 2 non-European populations from the 1000 Genomes Project: Yoruba Nigerians (YRI, N = 108) and Gujarati Indians (GIH, N = 103). We then estimated between-population $\hat{F}_{ST}$ as:

$$\hat{F}_{ST} = \frac{(\hat{p}_1 - \hat{p}_2)^2}{4\bar{p}(1 - \bar{p})}$$

where $\hat{p}_1$ and $\hat{p}_2$ are allele frequency estimates in the relevant pair of populations and $\bar{p} = (\hat{p}_1 + \hat{p}_2)/2$. We also used vcftools [92] to calculate Tajima's D, a metric of genetic diversity which takes on elevated values under certain evolutionary and demographic scenarios, including balancing selection, in 10 kb windows across the genome.

**Previous candidates for balancing selection.** If candidate SA variants experience strong balancing selection, they should disproportionately co-occur with previously identified candidates for balancing selection. We used 3 independent sets of candidate sites for balancing selection to investigate this possibility: (i) the dataset of Andrés and colleagues [62], which consists of 64 genes exhibiting elevated polymorphism (as determined using the Hudson–Kreitman–Aguadé test) and/or intermediate-frequency alleles across 19 African-American or 20 European-American individuals; (ii) the dataset of DeGiorgio and colleagues [64], which consists of 400 candidate genes exhibiting elevated $T_1$ or $T_2$ statistics among 9 European (CEU) and 9 African (YRI) individuals. $T_1$ or $T_2$ statistics quantify the likelihood that a genomic region exhibits levels of neutral polymorphism that are consistent with a linked balanced polymorphism; (iii) the dataset of Bitarello and colleagues [63], which consists of 1,859 candidate genes exhibiting elevated values of "non-central deviation" (NCD) statistics. NCD statistics also quantify the likelihood that given genomic regions are situated nearby a balanced polymorphism, using polymorphism data from 50 random individuals from 2 African (YRI; LWK) and European (GBR; TSI) populations and divergence data from a chimpanzee outgroup.

We assigned each site in the UK Biobank dataset to a gene using SnpEff [93] and categorised sites as candidates or non-candidates for balancing selection based on whether they were annotated as belonging to a candidate or non-candidate gene in each of the 3 aforementioned datasets.

**Statistical associations between metrics of sex-differential selection and balancing selection.** To test whether signals of sex differences in selection were associated with signals of balancing selection, we performed Spearman's rank correlations between alternative allele age (scaled by alternative allele frequency in the UK Biobank, to control for ascertainment bias) and each metric of sex differences in selection. For between-population $\hat{F}_{ST}$ and Tajima's D, we performed multiple linear regressions, with the relevant metric of sex differences in selection as the independent variable and MAF in the UK Biobank as a fixed-effect covariate (to control for ascertainment bias). For previous candidates for balancing selection, we performed multiple logistic regressions, where candidate/non-candidate status was the binary response variable, with the relevant metric of sex differences in selection as the independent variable and MAF in the UK Biobank as a fixed-effect covariate. In the case of regressions involving between-population $\hat{F}_{ST}$, $\hat{F}_{ST}$ was first log-transformed to meet normality assumptions. In the case of Tajima's D analyses (which are window-based rather than SNP-based), we averaged independent variables across 10 kb windows before performing regressions.

## Supporting information

**S1 Appendix. Supporting information.** Section A. Theoretical null distributions for $F_{ST}$ estimates. Section B. Hitchhiking estimates in between-sex $F_{ST}$. Section C. Defining upper bounds for excess heterozygosity in $F_{IS}$ estimates arising from SA selection. Section D. Polygenicity of

signals of sex differences in selection. Section E. Null model for unfolded reproductive $F_{ST}$. Section F. Correcting for sex-specific population structure. Section G. Sex differences in selection and the relation between $F_{ST}$ and MAF. Section H. Associations between metrics of sex differences in selection and MAF. Section I. Associations between metrics of sex differences in selection and candidates for balancing selection.
(PDF)

## Acknowledgments

We thank Tom Allison, Isobel Beasley, Isobel Booksmythe, Katja Kasimatis, and Max Reuter for helpful comments on previous versions of the manuscript. We thank the Monash Bioinformatics Platform for support.

## Author Contributions

**Conceptualization:** Filip Ruzicka, Luke Holman, Tim Connallon.

**Data curation:** Filip Ruzicka, Luke Holman.

**Formal analysis:** Filip Ruzicka, Luke Holman, Tim Connallon.

**Funding acquisition:** Tim Connallon.

**Investigation:** Filip Ruzicka, Luke Holman, Tim Connallon.

**Methodology:** Filip Ruzicka, Luke Holman, Tim Connallon.

**Project administration:** Filip Ruzicka, Tim Connallon.

**Visualization:** Filip Ruzicka, Luke Holman, Tim Connallon.

**Writing – original draft:** Filip Ruzicka, Tim Connallon.

**Writing – review & editing:** Filip Ruzicka, Luke Holman, Tim Connallon.

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
