## [Editor Report · Decision Letter 0]

7 Oct 2021

Dear Dr Ruzicka, 

Thank you for submitting your manuscript entitled "Polygenic signals of sexually antagonistic selection in contemporary human genomes" for consideration as a Research Article by PLOS Biology.

Your manuscript has now been evaluated by the PLOS Biology editorial staff, as well as by an academic editor with relevant expertise, and I'm writing to let you know that we would like to send your submission out for external peer review.

Once your full submission is complete, your paper will undergo a series of checks in preparation for peer review. Once your manuscript has passed the checks it will be sent out for review. 

If your manuscript has been previously reviewed at another journal, PLOS Biology is willing to work with those reviews in order to avoid re-starting the process. Submission of the previous reviews is entirely optional and our ability to use them effectively will depend on the willingness of the previous journal to confirm the content of the reports and share the reviewer identities. Please note that we reserve the right to invite additional reviewers if we consider that additional/independent reviewers are needed, although we aim to avoid this as far as possible. In our experience, working with previous reviews does save time. 

If you would like to send your previous reviewer reports to us, please specify this in the cover letter, mentioning the name of the previous journal and the manuscript ID the study was given, and include a point-by-point response to reviewers that details how you have or plan to address the reviewers' concerns. Please contact me at the email that can be found below my signature if you have questions. 

Please re-submit your manuscript within two working days, i.e. by Oct 11 2021 11:59PM.

Kind regards,

Roli Roberts

Roland Roberts

Senior Editor

PLOS Biology

rroberts@plos.org

---

## [Decision Letter · Decision Letter 1]

1 Dec 2021

Dear Dr Ruzicka,

Thank you for submitting your manuscript "Polygenic signals of sexually antagonistic selection in contemporary human genomes" for consideration as a Research Article at PLOS Biology. Your manuscript has been evaluated by the PLOS Biology editors, an Academic Editor with relevant expertise, and by four independent reviewers.

IMPORTANT: We'd like to apologise for an unfortunate glitch in our process. When we called in the additional metadata in preparation for peer review, our template letter invited you to include any previous reviews from other journals. You did so, correctly, but our system should have then alerted us to this fact, at which point we would have taken those reviews into account, potentially contacting the other journal, and then highlighting this fact to our reviewers. Unfortunately this alert failed, and your paper went out to new reviewers, but including your reviews and rebuttal. You will see that most of the reviewers nevertheless acknowledge this fact, and have commented accordingly. Overall, we hope that this process will prove useful, despite not providing the expedited journey that we intended. I'll apologise similarly to the reviewers for the confusion.

When I described the situation to the Academic Editor, s/he said "I hope that you can convey that [the required revision] is at the milder end of the spectrum. It has been reviewed already, and several of the new reviewers noted that they were therefore reluctant to require yet more analyses. The suggested analyses make sense to me, and would be helpful, but I think that the main issue is to present the work more cautiously, which should be straightforward." I hope that this additional advice will be helpful when deciding how to revise your manuscript.

In light of the reviews (below), we will not be able to accept the current version of the manuscript, but we would welcome re-submission of a much-revised version that takes into account the reviewers' comments. We cannot make any decision about publication until we have seen the revised manuscript and your response to the reviewers' comments. Your revised manuscript is also likely to be sent for further evaluation by the reviewers.

We expect to receive your revised manuscript within 3 months. 

**IMPORTANT - SUBMITTING YOUR REVISION**

*Re-submission Checklist*

*Published Peer Review*

*PLOS Data Policy*

*Blot and Gel Data Policy*

Sincerely,

Roli Roberts

Roland Roberts

Senior Editor

PLOS Biology

rroberts@plos.org

REVIEWERS' COMMENTS:

Reviewer #1:

[identifies himself as Daniel Berner]

Comments on PBIOLOGY-D-21-02588R1, 'Polygenic signals of sexually antagonistic selection in contemporary human genomes'

I have digested this study as a new reviewer, and to save time, I have haphazardly but not systematically checked how the authors addressed previous reviewer criticism.

This study uses a very large human data set, including whole-genome genotype data and information on offspring number, to look for genome-wide signatures of sexually antagonistic (SA) selection. The key analyses include the evaluation of the observed distribution of intersex differentiation (Fst) against null expectations, the association of Fst with MAF, the location of candidate SA SNPs in (non)functional genomic regions, and looking for evidence of balancing selection at candidate SA SNPs. The study performs quite rigorous data filtering to reduce the risk of detecting spurious SA signals, and the theoretical framework underlying their analyses is very carefully developed in my view. The key findings include a collective (but of course not SNP-level) pattern consistent with SA selection on both survival and reproductive output emerging from several analyses, and that SA candidate loci are enriched for functional sites. The study finds no evidence, however, that SA candidate loci are under balancing selection.

The quest for demonstrating SA selection is currently an active field in biology, so the study is certainly topical. Moreover, I feel that methodologically, the study is quite rigorous and in many aspects goes beyond what has previously been presented in the field. Of course there have been (more or less problematic) previous attempts to detect SA selection in humans, so this work is not addressing an entirely novel question or producing earth-shattering results. Nevertheless, the overall quality of the investigation and evidence seems outstanding to me, and I feel it is a strength, not a weakness, to revisit important existing research problems with new data and methods. Also, the writing is very clear and analyses are well justified and explained; I really enjoyed reading this manuscript and feel the investigation is well suited for publication in PLoS Biology.

I consider this work in very good shape (it is a revision, after all) and have only relatively minor comments. Perhaps the most serious would be to consider testing for elevated genetic diversity around the SA candidates, as a complementation to the balancing selection analysis. Here the details, not sorted by relevance:

1) Line 89 (and L139-140 etc): I was first puzzled by this association between Fst and MAF being taken as evidence of SA selection, because for the allele frequency difference, a positive correlation to MAF exists under pure neutrality (because low-MAF polymorphisms cannot yield high differentiation; Roesti et al. 2012 BMC Evol Biol). I thus performed a brief simulation experiment on the association between differentiation and MAF in the absence of SA selection. This indicated that indeed the raw intersex allele frequency difference is correlated positively to MAF, but Fst is not. Given that the Fst-MAF correlation is just one of the signals taken as evidence of SA selection by the authors, and that its theoretical basis is well developed in the Methods section, I trust this conclusion, although it goes somewhat beyond my direct intuition.

2) L114: This passage makes me wonder about loci that may have antagonistic fitness effects within a sex across the life cycle. E.g., loci that may increase female survival but reduce female fertility. Is this possibility addressed in the paper (I may have overlooked)?

3) L189 and elsewhere: I think in line with what professional statisticians recommend, the term 'statistically significant' should be discarded (e.g., Wasserstein et al. 2019, American Statistician). This dichotomous label is not needed; can simply give the P-value, or even better, the parameter estimates along with their confidence intervals.

4) L296-302: I think a signature consistent with balancing selection would be elevated genetic diversity in the close (i.e., tightly physically linked) vicinity of the candidate SA polymorphisms. I encourage the authors to explore this. For instance, take your best few hundred or so SA candidate SNPs, and retrieve all polymorphisms occurring within windows of, say, 4 kb around them. Then do the same for a large number of non-SA-candidate SNPs. A prediction would be that around the SA candidate SNPs, genetic variation is elevated. Not sure what would be the best diversity metric, but Haenel et al. (2019, Evolution Letters) suggest that the density of high-MAF SNPs is a particularly sensitive diversity measure.

5) L351-355: You here argue that sample size could be an issue, but how large are the Biobank Japan and Finngen data bases?

6) L424: These (or analogous) predictions for mapping artifacts have also been described in previous studies, which could perhaps be acknowledged:

https://doi.org/10.1186/1745-6150-7-17

https ://doi.org/10.1111/1755-0998.12613

https ://doi.org/10.3390/genes 10040320

https ://doi.org/10.1111/mec.15255

7) L439, ‚eliminate mapping errors'. Perhaps tone this down slightly; using such a threshold-based approach, it is unlikely that errors are completely eliminated, as some errors may just pass below the detection criterion.

8) L473: Put 'proxy' in singular.

9) Figure 2A-C: How about adding a LOESS smoother to the observed data? Also, in B, is there an explanation as to why the permuted data also seem to show an excess of SNPs in the top Fst quantiles?

10) L902 and 915: I generally think that a lot of significance testing we report is unnecessary. Here, for instance, the histograms convey the full information, no need for P values in my view.

Reviewer #2:

The work has been reviewed before by three reviewers and so I will comment on the responses to these first. 

1. Generally, the authors have responded well to the comments of Reviewer #1 and Reviewer #2.

2. I agree with the previous reviewers regarding the fact that the mixed-model association FST values should be the focus of the manuscript. It would streamline the presentation, especially of the results, which remain hard to follow, and as the authors state in their response to Reviewer #3, it is these estimates that have the best chance to control for population artefacts (though see below). 

3. I do not feel they have addressed the concern of Reviewer #3 in their final major point regarding the breakdown of the data into signals of viability and reporductive fitness.

4. The authors need to conclusively demonstrate why there is a differenc between the theoretical and empirical nulls, rather than providing conjecture that it could stem from deviations from HWE.

I am loathe to propose further work given that the manuscript has already been under review. However, I strongly feel that there are some key issues that remain unresolved. The authors claim that this is "the first study to present unambiguous signals of sex-differential selection in human genomes" and they claim to show wide-spread, polygenic, sex-differential selection that has a major impact on genomic variation. This is a fairly bold claim and as such requires stronger evidence than I feel has been presented here.

1. I am concerned that there is the potential for sample ascertainment to give extensive bias. The UK Biobank is a healthier sample than the general UK population and the sample is only 46% male. So what causes females to be more likely to register to the UK Biobank? Are there any traits with a genetic basis that could be associated with this ascertainemnt, which then translates ascertainment into sex biases at the frequencies of different alleles. I suspect so, and there is nothing one can do about it. This issue goes beyond discussing this as a potential caveat. As the authors make extrodinary claims - that sexual antagonism maintains variation at a substantial number of locations across the genome (i.e. it is polygenic) - there is a requirement for more evidence then a single association study. I would be more convinced if they replicated the top associations within another biobank. I also feel they need to show that ascertainment of individuals could not cause the results observed here. They also need to control for age-at-enrolment within their analyses.

2. Does between-sex FST increase with the age group of the participants? Women have a higher life expectancy and so will be over-represented in the older age classes. LRS, as measured by number of children, has varied greatly due to cultural factors over the past decades, with family sizes generally becomming smaller over time. As women simply survive longer and are more likely to enter the study in later-life, could these patterns not simlply represent cultural changes across generations linked to viability selection?

3. The authors should use LDScore regression to calculate the correlation between LRS and FST from the mixed model summary statistics. Also as a way to test for enrichment using annotation groups. This would be far more robust (controlling for intercept terms which may reflect population stratification) than the t-statistics presented here.

4. The null model for LRS should not be based on randomised trait values within each sex but rather based on a simulated heritable trait with equal mean and variance across the sexes and genetic correlation = 1.

5. Please also fit PCs of the chromosomes as fixed effects when calculating the LOCO regression coefficients. BoltLMM controls for general population stratification on chromosomes other than the focal, but not the one in which the marker effect size is being estimated.

Reviewer #3:

Ruzicka et al. develop a suit of methods to test for contemporary sexually-antagonistic (SA) viability selection—using allele frequency differences between males and females in a large cohort—and reproductive selection—using the number of children reported for said individuals. They apply these methods to the UK Biobank and reject a null model of zero SA selection in this cohort.

The paper has many notable strengths: Readers will benefit from the inclusion of a theoretical expectation and the permutation-based null alongside it; the control for population structure is novel (though the authors, unfortunately, do not ask whether it successfully removed some of the stratification due to participation bias that Pirastu et al., Nature Genetics 2021 reported); the estimates of SA selection that they derive will be useful for the community and the analysis is one of the most thorough I have seen in this space. The breakdown into the fitness components is cool.

At the same time, key choices made by the authors need to be revisited in order for the manuscript to merit publication. Most importantly, these include presenting the most conservative version of their analyses in the main text rather than in the supplement—in particular setting a proper null expectation and control for LD in all analyses. If these issues, as detailed below, can be solved, I think the work would make for a valuable contribution to the literature.

The authors do well to clarify that they are not estimating the strength of sexually antagonistic selection (SAS) or its importance in shaping patterns of genetic variation. Rather, they set more modest goals of testing a null hypothesis of no contemporary SA selection, and pointing to the likeliest candidate targets of SAS. Given this fact, however, I would encourage the authors to show an LD-pruned and population structure-aware version of their tests as main figures and text. The current presentation of the results in the main text is anti-conservative, as it is inflated by both factors.

At several points along the paper, the authors seem to implicitly assume "all or nothing" theoretical models or interpretation that weaken their analyses and conclusions somewhat. One example is in the relationship reported for minor allele frequency and Fst (around line 212)—a part I might, as the previous cycle's reviewer 1 had suggested, either remove or further flesh out possible interpretations and caveats to the conclusion. An example caveat is ascertainment bias. Do the authors observe similar relationships with allele frequency in a different sample (with relatively diverged ancestry)?

Another example is in the overarching assumption that alleles are either experience the same selective pressures in males and females or are subject to SA selection. What about the (seemingly more likely, agnostically) case of sex-specific selection, in which selective effects differ to some unknown extent between the sexes? 

Yet another example of the "all or nothing" interpretive approach is in a section entitled "Evolutionary Analysis of candidate SA SNPs" (lines 282-302. The authors test for a correlation between measures of balancing selection and between-sex Fst. We might, at best, learn that there is no evidence for a very strong hypothesis of a monotonic relationship between contemporary SA selection and long-term balancing selection—but this is a rather weak statement, and one that the authors still need to put in substantial work to establish. (Because, as in other parts of the paper, there is no proper control for LD or consideration of estimation noise).

The authors also innovate in their control for population structure stratification. They replace Fst with the test statistic Lst, the log-odds ratio of an allele being carried by a male relative to a female, beyond what can be explained by the main axes of genetic ancestry (i.e., by including principal components of the genotype matrix as covariates). Is this approach enough to remove some of the stratification reported by Pirastu et al.? Or does everything that follows boil down to differences in interpretation of the same signals, where Ruzicka et al. interpret them as SA selection and Pirastu et al. interpret them as artefacts due to participation bias? Readers would benefit from some discussion on this.

The analysis of correlation with GWAS effect estimates seems invalid to me, because of the same two familiar reasons: It does not set up a null, empirical or otherwise, and does control for LD in any way. For the latter, if complex traits like testosterone are the trait of interest, genetic correlations can be tested for rather than raw correlations. 

Speaking of T, the observation "Loci associated with T had lower than average gametic Fst" needs to be fleshed out: There are no statistics to back it up, and the authors do not suggest what it might mean. This remains especially unclear to me given that variation in testosterone levels have highly distinct genetic bases in the two sexes (Flynn et al., Eur J Hum Genetics 2021; Sinnott-Armstrong, Naqvi et al., eLife 2021).

Minor comments:

- Can the authors cite sources supporting the choice of cutoff age of 45 for child bearing / having? Also, does the UKB questionnaire ask only about biological children?

- In the UKB (an order of a million chromosomes sampled at each site), a 1% cutoff means essentially no rare variation will be observed, further weakening the conclusion of the analysis on the relationship between minor allele frequency and SA selection.

Reviewer #4:

This review is incomplete (due to a mistake on the reviewer's part about timing of the deadline), so some aspects of the how the methods yielded the results observed may not have been fully considered by the reviewer.

General comment (written quickly this morning):

I really liked this paper -- I think it shows a conservative approach for testing for the effect of sexually antagonistic selection in humans, which is an area of considerable interest. The authors seem to have gone to extensive lengths to try to control a range of sources of error, considering other previous analyses and implementing controls from them. If anything, I think some of these controls may cause an underestimation of the effects. As with anything highly polygenic and driven by subtle allele frequency changes, I am still not sure there isn't some unconsidered variable driving these observations, but I have been mulling it over for a few weeks now and nothing obvious would make sense. Thus, I think it's a good candidate for PLOS Biology. Well written and clear.

Other comments organized from previous notes:

Major comments: 

1. Any explanation for why the difference between observed and theoretical is so much larger than observed and null for panel 2E? The other ones look like good agreement between theoretical and null distributions. This one is looking at the difference between adult and offspring allele frequencies in male vs. difference in females. I was surprised by the differences in this figure and felt like I would have liked some more explanation about why that might occur.

Minor comments:

Line 159: Is it really true that intermediate frequency alleles inflate FST_hat more than low frequency alleles under drift? I thought the whole point of FST is that it's representing drift irrespective of allele frequency. Not sure about this particular point.

Line 174: I can appreciate that to be conservative and make it harder to detect SA selection, you need to adopt these filtering criteria, but couldn't differences in missing rates be caused by an indel that has been favoured in one sex? Would this increase chance of false negatives? Or are these really egregious bionformatic errors? I guess the method described on line 425-441 is thought to only exclude loci under implausibly strong SA selection? (given human mortality data)?

Figure 2: I can appreciate that you'd like to maintain the same colour scheme for panels D-F as in A-C, but I didn't read the caption carefully at first and I thought that the open bars were somehow representing the null and coloured bars representing the data. If you could somehow more clearly emphasize on the Figure itself that these are representing the difference between observed - null and observed - theoretical, that would help. I think making both hollow and filled bars the same colour would help me notice this more easily. Maybe add a legned with a hollow box and a filled box?

Line 205: I'm not sure that this is evidence about polygenic. If we knew nothing about linkage, it could be possible that all the enriched high-FST SNPs occurred in a single block of highly linked alleles with one causal allele under strong selection that simply wasn't quite enough to get above the bonferroni. 

Line 262: what are the effect sizes here? With these types of tests, significant p-values can be found with very small effects, as the sample size is very large. Can you provide some estimate of odds ratio or something? (how much more likely is a high FST to be found in a genic vs. non-genic region).

Line 269: It wasn't totally clear to me what was being done here without going to the methods, so a little more explanation would help. What is the assocation that is being tested on line 272? Is this testing somehow whether the high FST SNPs are enriched among the GWAS candidate loci? It's explained clearly on line 555 but would be easier to understand with a bit more detail in the results where it's first introduced.

Line 280: Interesting that loci affecting testosterone had lower than average FST -- but I guess maybe that's a conflict that has already been resolved to the sex chromosomes?

Line 368: Interesting that there was no signature of balancing selection on these loci -- reasons outlined here are appropriate.

---

## [Decision Letter · Decision Letter 2]

6 Jul 2022

Dear Dr Ruzicka,

Thank you for your patience while we considered your revised manuscript "Polygenic signals of sex differences in selection in contemporary humans" for publication as a Research Article at PLOS Biology. This revised version of your manuscript has been evaluated by the PLOS Biology editors, the Academic Editor and two of the original reviewers.

Based on the reviews , we are likely to accept this manuscript for publication, provided you satisfactorily address the remaining points raised by the reviewers. Please also make sure to address the following data and other policy-related requests.

IMPORTANT:

a) Please make the title more declarative. We suggest "UK Biobank data reveal polygenic signals of sex differences in selection in contemporary humans" - this contains an active verb and mentions the source of the data.

b) Please attend to the remaining modest requests from reviewer #3.

c) Please address my Data Policy requests below; specifically, we need you to supply the numerical values underlying Figs 2ABCDEF, 3BC, 4ABCDEF, 5ABC, 6ABCDEFGH, SD1, SD2, SF1, SF2, three unnumbered Figs in SG, SH1, SI1, SI2. I note that your Github depositions are currently empty; please complete these so that we can check policy compliance. In addition, we need a citeable, permanent record of the data, e.g. in Zenodo.

d) Please also cite the location of the data clearly in each main and supplementary Fig legend, e.g. “The data and code needed to generate this Figure can be found in https://github.com/filruzicka/polygenic_SA_selection_in_the_UK_biobank
https://github.com/lukeholman/UKBB_LDSC and https://zenodo.org/record/XXXXXX.”

We expect to receive your revised manuscript within two weeks. 

*Published Peer Review History*

*Press*

Sincerely,

Roli Roberts

Roland Roberts, PhD

Senior Editor,

rroberts@plos.org,

PLOS Biology

DATA POLICY:

Regardless of the method selected, please ensure that you provide the individual numerical values that underlie the summary data displayed in the following figure panels as they are essential for readers to assess your analysis and to reproduce it: Figs 2ABCDEF, 3BC, 4ABCDEF, 5ABC, 6ABCDEFGH, SD1, SD2, SF1, SF2, three unnumbered Figs in SG, SH1, SI1, SI2. NOTE: the numerical data provided should include all replicates AND the way in which the plotted mean and errors were derived (it should not present only the mean/average values).

DATA NOT SHOWN?

REVIEWERS' COMMENTS:

Reviewer #1:

I have now gone through the revision (PBIOLOGY-D-21-02588R2) of the manuscript I reviewed earlier. I have no further criticism; I think the authors have made a very careful and extensive revision suited for publication. I feel the paper will make a fine contribution to PLoS Biol!

Reviewer #3:

[identifies himself as Arbel Harpak]

Overall, I think the authors did an excellent job at addressing reviewers' comments seriously and thoroughly. I include a few comments / questions I am curious about below, but strongly believe that at this point the authors have earned the right to leave a small fraction of (now seven!) reviewers' musings unattended… 

1. I did not understand the response to my last comment asking why alleles with maf <1% were excluded. I think that in practice examining only very common alleles may give a warped view of selection.

2. I remain curious about the weird correlation of SD-selection statistics with testosterone (or, in the current version, lack of correlation). A seemingly very related 2022 bioRxiv preprint by Zhu et al. examined the relationship between sex differences in genetic effects on traits and male-female Fst. Only one trait repeatedly (in three datasets; though much smaller than the UKB) came up as slightly significant---testosterone. Can the authors comment on whether or not this is discrepant with their results, and why? On that note, perhaps some of the GWAS analyzed in Fig. 5C should be repeated with a sex-stratified GWAS. In particular extremes like testosterone where genetic effects on the trait are highly sex-specific.

3. I appreciate the efforts by the authors to make the language and the presentation more careful, but I think there's still some room for improvement. One example: The legend of figure 5 says that "positive correlations indicate that "a trait is more beneficial in females than in males". This is both ill-defined (I believe the authors you mean higher trait values are favored) and overstates. The statement can be made about the benefit of alleles associated with the trait; but given plausible complications, such as pleiotropy, recruitment biases… I think any claim from a single-trait test that does not consider correlations between traits, levels of background selection, recruitment or other modes of population stratification is at best suggestive about the focal trait being under selection. The authors should try and be extra cautious when it comes to claim about sex-specific selection, especially when they themselves fully acknowledge the looming caveat of study recruitment biases.

---

## [Editor Report · Decision Letter 3]

27 Jul 2022

Dear Dr. Ruzicka,

Thank you for the submission of your revised Research Article "Polygenic signals of sex differences in selection in humans from the UK Biobank" for publication in PLOS Biology. On behalf of my colleagues and the Academic Editor, Nick Barton, I am pleased to say that we can in principle accept your manuscript for publication, provided you address any remaining formatting and reporting issues. These will be detailed in an email you should receive within 2-3 business days from our colleagues in the journal operations team; no action is required from you until then. Please note that we will not be able to formally accept your manuscript and schedule it for publication until you have completed any requested changes.

PRESS

Sincerely, 

Paula Jauregui on behalf of 

Senior Editor

PLOS Biology

rroberts@plos.org